



# Seasonal variations in photooxidant formation and light absorption in aqueous extracts of ambient particles

Lan Ma[1a], Reed Worland[1b], Laura Heinlein[1], Chrystal Guzman[1c], Wenqing Jiang[2], Christopher Niedek[2], Keith J. Bein[3], Qi Zhang[2], Cort Anastasio[1]

[1]Department of Land, Air and Water Resources, University of California, Davis, One Shields Avenue, Davis, CA 95616-8627, USA
[2]Department of Environmental Toxicology, University of California, Davis, One Shields Avenue, Davis, CA 95616-8627, USA
[3]Center for Health and the Environment, University of California, Davis, One Shields Avenue, Davis, CA 95616-8627, USA
[a]Now at: SGS-CSTC Standards Technical Services Co.,Ltd. Hangzhou Branch, Hangzhou, Zhejiang Province, 310052, China
[b]Now at: Department of Chemistry, University of Washington, WA, USA
[c]Now at: Department of Pharmacology, University of Washington, WA, USA

*Correspondence to:* Cort Anastasio (canastasio@ucdavis.edu)

**Abstract.** Atmospheric waters – including fog/cloud drops and aerosol liquid water – are important sites for the transformations of atmospheric species, largely through reactions with photoformed oxidants such as hydroxyl radical ($^{\bullet}$OH), singlet molecular oxygen ($^1O_2^*$), and oxidizing triplet excited states of organic matter ($^3C^*$). Despite this, there are few measurements of these photooxidants, especially in extracts of ambient particles, and very little information about how oxidant levels vary with season or particle type. To address this gap, we collected ambient $PM_{2.5}$ from Davis, California over the course of a year and measured photooxidant concentrations in dilute aqueous extracts of the particles. We categorized samples into four groups: Winter & Spring (Win-Spr), Summer & Fall (Sum-Fall) without wildfire influence, fresh biomass burning (FBB), and aged biomass burning (ABB). FBB contains significant amounts of brown carbon (BrC) from wildfires, and the highest mass absorption coefficients (MAC) normalized by dissolved organic carbon, with an average ($\pm 1 \sigma$) value of 3.3 ($\pm 0.4$) $m^2$ (g C)$^{-1}$ at 300 nm. Win-Spr and ABB have similar MAC averages, 1.9 ($\pm 0.4$) and 1.5 ($\pm 0.3$) $m^2$ (g C)$^{-1}$, respectively, while Sum-Fall has the lowest MAC$_{DOC}$ (0.65 ($\pm 0.19$) $m^2$ (g C)$^{-1}$). $^{\bullet}$OH concentrations in extracts range from (0.2-4.7) $\times 10^{-15}$ M and generally increase with concentration of dissolved organic carbon (DOC), although this might be because DOC is a proxy for extract concentration. The average quantum yield for $^{\bullet}$OH formation ($\Phi_{OH}$) across all sample types is 3.7 ($\pm 2.4$) %, with no statistical difference among sample types. $^1O_2^*$ concentrations have a range of (0.7-45) $\times 10^{-13}$ M, exhibiting a good linearity with DOC that is independent of sample type ($R^2 = 0.93$). Fresh BB samples have the highest [$^1O_2^*$] but the lowest average $\Phi_{1O2^*}$, while Sum-Fall samples are the opposite. $\Phi_{1O2^*}$ is negatively correlated with MAC$_{DOC}$, indicating that less light-absorbing samples form $^1O_2^*$ more efficiently. We quantified $^3C^*$ concentrations with two triplet probes: syringol (SYR), which captures both strongly and weakly oxidizing triplets, and (phenylthio)acetic acid (PTA), which is only sensitive to strongly oxidizing triplets. Concentrations of $^3C^*$ are in the range of (0.03 – 7.9) $\times 10^{-13}$ M and linearly increase with DOC ($R^2 = 0.85$ for SYR and $R^2 = 0.80$ for PTA); this relationship for [$^3C^*$]$_{SYR}$ is independent of sample type. The average ratio of [$^3C^*$]$_{PTA}$/[$^3C^*$]$_{SYR}$ is 0.58 ($\pm 0.38$), indicating that roughly 60% of oxidizing triplets are strongly oxidizing. Win-Spr samples have the highest fraction of strongly oxidizing $^3C^*$, with an average of 86 ($\pm 43$)%. $\Phi_{3C^*,SYR}$ is in the range of (0.6-8.8) %, with an average value, 3.3 ($\pm 1.9$)%, two times higher than $\Phi_{3C^*,PTA}$. FBB has the lowest average $\Phi_{3C^*}$, while the aging process tends to enhance $\Phi_{3C^*}$, as well as $\Phi_{1O2^*}$.

To estimate photooxidant concentrations in particle water, we extrapolate the photooxidant kinetics in our dilute particle extracts to aerosol liquid water (ALW) conditions of 1 µg PM/µg H$_2$O for each sample type. The estimated ALW $^{\bullet}$OH concentration is 7 $\times 10^{-15}$ M when including mass transport of gas-phase $^{\bullet}$OH to the particles. $^1O_2^*$ and $^3C^*$ concentrations in ALW have ranges of



$(0.6 - 7) \times 10^{-12}$ M and $(0.08 - 1) \times 10^{-12}$ M, respectively. In the Win-Spr and Sum-Fall samples, photooxidant concentrations increase significantly from lab particle extracts to ALW, while the changes for the FBB and ABB samples are minor. The small increases in $^1O_2^*$ and $^3C^*$ from extract to ALW for the biomass burning particles are likely due to the high amounts of organic compounds in the extracts, which lead to strong quenching of these oxidants even under our dilute conditions. Compared to the photooxidant concentration estimates in Kaur et al. (2019), our updated ALW estimates show higher $^\bullet OH$ (by roughly a factor of

10), higher $^3C^*$ (by factors of 1-5) and lower $^1O_2^*$ concentrations (by factors of 20-100). Our results indicate that $^3C^*$ and $^1O_2^*$ in ALW dominate the processing of organic compounds that react quickly with these oxidants (such as phenols and furans, respectively), while $^\bullet OH$ is more important for less reactive organics.

## 1 Introduction

Atmospheric waters, including fog/cloud drops and liquid water on aerosol particles, are important media for photochemical

transformations of chemical species (Herrmann et al., 2010, 2015). These include formation of aqueous secondary organic aerosol (aqSOA), formation and photobleaching of brown carbon (BrC), oxidation of reduced sulfur, and aerosol aging (Ervens, 2018; Ervens et al., 2011; Gilardoni et al., 2016; Laskin et al., 2015; McNeill, 2015; Seinfeld and Pandis, 2016; Wang et al., 2016; Zhao et al., 2015). Many of these processes are driven by photochemically generated oxidants, including hydroxyl radical ($^\bullet OH$), triplet excited states of organic matter ($^3C^*$), and singlet molecular oxygen ($^1O_2^*$) (Ervens et al., 2014; Finlayson-Pitts and Pitts, 2000;

He et al., 2013; Herrmann, 2003; Kaur et al., 2019; Lim et al., 2010).

Hydroxyl radical ($^\bullet OH$), the best studied aqueous oxidant in the atmosphere, is highly reactive with most reduced species but has a relatively low abundance compared to $^3C^*$ and $^1O_2^*$. Concentrations of $^\bullet OH$ in fog and cloud waters, as well as aqueous extracts of ambient particles and lab-generated secondary organic aerosol, are typically $10^{-17}$ to $10^{-15}$ M (Anastasio and McGregor, 2001;

Arakaki et al., 2013; Dorfman and Adams, 1973; Kaur and Anastasio, 2017; Kaur et al., 2019; Manfrin et al., 2019; Tilgner and Herrmann, 2018). Sources of $^\bullet OH$ in the aqueous phase include mass transfer from the gas phase, Fenton or Fenton-like reactions of reduced metals with hydrogen peroxide, and photolysis of nitrate, nitrite, iron complexes, hydrogen peroxide, and organic hydroperoxides (Badali et al., 2015; Herrmann et al., 2010; Tilgner and Herrmann, 2018; Tong et al., 2016). Additionally, organic compounds in atmospheric waters can affect $^\bullet OH$ production. For example, the interaction of humic-like substances (HULIS) or

SOA with Fe(II) can enhance or suppress $^\bullet OH$ formation (Baba et al., 2015; Gonzalez et al., 2017; Hems and Abbatt, 2018; Tong et al., 2016; Zuo and Hoigne, 1992). This suggests that seasonal variations in particle composition (e.g., SOA and Fe) can affect $^\bullet OH$ kinetics, as reported recently for $^\bullet OH$ photoproduction in extracts of particulate matter (PM) from Colorado: winter $^\bullet OH$ originated from nitrate photolysis, while summer $^\bullet OH$ was more linked to soluble iron (Leresche et al., 2021). But little is known about how $^\bullet OH$ concentrations in particles vary with season or among particle types.


Triplet excited states ($^3CDOM^*$) are formed when organic chromophores (i.e., brown carbon (BrC)) absorb sunlight and are promoted to a higher energy state (McNeill and Canonica, 2016). Oxidizing triplets ($^3C^*$), i.e., the subset of triplets that have high reduction potentials, are effective oxidants, reacting with phenols and biogenic volatile compounds to form SOA and BrC, and oxidizing bisulfite to sulfate (González Palacios et al., 2016; Monge et al., 2012; Rossignol et al., 2014; Smith et al., 2014; Wang

et al., 2020b; Yu et al., 2014). For compounds (like phenols) that react rapidly with triplets, $^3C^*$ can be as important an oxidant as $^\bullet OH$ in cloud and fog drops, where oxidizing triplet concentrations are $10^{-15}$-$10^{-13}$ M (Kaur and Anastasio, 2018; Kaur et al., 2019; Ma et al., 2021; Smith et al., 2015). Moreover, triplet concentrations are estimated to be enhanced by one or two orders of



magnitude in aerosol liquid water (Kaur et al., 2019; Ma et al., 2023a). The ability of dissolved organic matter (DOM) to form $^3C*$ depends on its composition. In surface waters, quantum yields of $^3C*$ are higher for organic compounds with lower average

molecular weights and lower aromaticity (Berg et al., 2019; Maizel and Remucal, 2017; McCabe and Arnold, 2017, 2018; Mckay et al., 2017). However, little is known about how $^3C*$ formation in atmospheric waters depends on BrC characteristics or season.

The final oxidant we consider, singlet molecular oxygen ($^1O_2*$), is formed when triplet excited states transfer energy to dissolved molecular oxygen. $^1O_2*$ reacts rapidly with electron-rich compounds such as furans, polycyclic aromatic hydrocarbons, some

amino acids, and substituted alkenes (Gollnick and Griesbeck, 1985; McGregor and Anastasio, 2001; Richards-Henderson et al., 2015; Wilkinson et al., 1995; Zeinali et al., 2019). $^1O_2*$ concentrations in fog and cloud waters and aqueous particle extracts are the highest of the three oxidants, in the  range of $10^{-14}$-$10^{-12}$ M (Bogler et al., 2022; Kaur and Anastasio, 2017; Kaur et al., 2019; Leresche et al., 2021; Manfrin et al., 2019). Dissolved black carbon also can produce $^1O_2*$, resulting in concentrations on the order of $10^{-12}$ M (Li et al., 2019). Though $^1O_2*$ is not as reactive as $^3C*$ and $^•OH$, its concentration increases by orders of magnitude

when moving from dilute cloud/fog conditions towards the more concentrated conditions of aerosol liquid water (Kaur et al., 2019; Ma et al., 2023a). Since $^1O_2*$ is born from $^3CDOM*$, these two oxidants are tightly linked. For example, in surface waters the quantum yield of $^1O_2*$ ($\Phi_{1O2*}$) is also higher in samples with lower average molecular weight DOM, as seen for $^3C*$ (Berg et al., 2019; Maizel and Remucal, 2017; Ossola et al., 2021; Wang et al., 2020a). Some studies on the seasonal trend of $\Phi_{1O2*}$ in surface waters hypothesized that summer samples where photodegradation is more rapid have higher $\Phi_{1O2*}$ based on DOM

photodegradation increasing $^1O_2*$ quantum yields (McCabe and Arnold, 2016; Ossola et al., 2021; Sharpless et al., 2014). However, there are differences in singlet oxygen generation and concentrations between surface and atmospheric waters. For example, while ozonation and photodegradation of DOM enhances $\Phi_{1O2*}$ in surface waters, photodegradation of aqueous particle extracts has no significant effect on $\Phi_{1O2*}$ (Leresche et al., 2019, 2021; Sharpless et al., 2014). In addition, we know very little about the seasonality of $^1O_2*$ concentrations in particles or how this oxidant varies between particle types.


Although $^•OH$, $^3C*$, and $^1O_2*$ are important in the transformation of atmospheric species, there are few measurements of these photooxidants in atmospheric condensed phases, especially in extracts of ambient particles. In addition, very little is known about seasonal variations in these oxidant concentrations and kinetics. To address this gap, we collected $PM_{2.5}$ from November 2019 to October 2020 in Davis CA, extracted them in water, and measured light absorption and photooxidant formation. This period

included four main types of samples: winter samples influenced by residential wood combustion and high humidity, summer samples impacted by nearby wildfires (i.e., fresh biomass burning (BB) particles), summer samples impacted by more distant wildfires (i.e., aged BB particles), and spring/summer samples with little to no biomass burning. We measured photooxidant concentrations ($^•OH$, $^1O_2*$, $^3C*$) in water extracts of the particles, and investigated how photooxidant formation depends on particle type, optical properties, and biomass burning influence. Finally, we extrapolated our dilute extract results to predict photooxidant

concentrations in aerosol liquid water (ALW) and assessed the importance of photooxidants in processing particulate organic compounds.

## 2 Experimental methods

### 2.1 Chemicals

Furfuryl alcohol (FFA, 98%), benzoic acid (BA, ≥ 99.5%), $p$-hydroxybenzoic acid ($p$-HBA, 99%), (phenylthio)acetic acid (PTA,

96%), syringol (SYR, 99%), 3,4-dimethoxybenzaldehyde (DMB, 99%), and deuterium oxide ($D_2O$, 99.9% D-atom) were received



from Millipore Sigma. All chemical solutions and particulate matter extracts were prepared using air-saturated ultrapure water (Milli-Q water) from a Milli-Q Advantage A10 system (Millipore; ≥18.2 MΩ cm) that was pretreated with a Barnstead activated carbon cartridge.

### 2.2 Particle collection and extraction

More detailed descriptions of sampling and extraction procedures are provided in Ma et al. (2023a) and are only briefly discussed here. Fine particle (PM$_{2.5}$) sampling was conducted from November 2019 to October 2020 on the roof of Ghausi Hall on the University of California, Davis campus. Winter in Davis is humid and sometimes foggy, and the air quality is often impacted by residential wood combustion, while Davis in summer is hot and dry. During the summer of 2020, several severe wildfires occurred in Northern California and Oregon, including the largest wildfires in the recorded history of California: the August complex (size:

4179 km$^2$), LNU Lightning complex (1605 km$^2$), and SCU lightning complex (1470 km$^2$) (https://www.fire.ca.gov/incidents/2020; last access: 15 July 2022). These fires caused extremely heavy air pollution in Davis with daily PM$_{2.5}$ concentrations sometimes exceeding 80 µg m$^{-3}$ (https://www.arb.ca.gov/aqmis2/aqmis2.php, last access: 20 June 2022). Particles were collected with a high-volume sampler containing a PM$_{10}$ inlet (Graseby Andersen) and two offset, slotted impactor plates (Tisch Environmental, Inc., 230 series) to remove particles larger than 2.5 µm. PM$_{2.5}$ was collected onto pre-cleaned Teflon-coated borosilicate glass microfiber

filters (Pall Corporation, EmFab$^{TM}$ filters, 8 in. × 10 in.) and stored at −20 °C immediately after collection. The sampling duration was either 24 hr or up to a week (Table S1). The sampling campaign was paused from March to June 2020 because of COVID-related restrictions on campus activities.

To prepare particulate matter extracts (PMEs), filters were cut into 2 cm × 2 cm squares, and then extracted with 1.0 mL Milli-Q

water by shaking for 4 h in the dark. The extracts from the same filter were combined, filtered (0.22 µm PTFE; Pall), and adjusted to pH 4.2 by sulfuric acid to mimic the acidity of winter particle water in the Central Valley of California (Parworth et al., 2017). The acidity of extracts was measured by a pH microelectrode (MI-414 series, protected tip; Microelectrodes, Inc.). PMEs were flash-frozen in liquid nitrogen immediately after preparation and were later thawed on the day of the experiment. Particle mass extracted was determined by weighing filter squares before and after extraction with a microbalance (M2P, Sartorius); the extracted

mass is an upper bound because we cannot account for insoluble material that is extracted from the square but removed by subsequent filtration. UV-Vis spectra of PMEs were measured with a Shimadzu UV-2501PC spectrophotometer. Dissolved organic carbon (DOC) and major ions were measured by a total organic carbon analyzer (TOC-VCPH, Shimadzu) and ion chromatographs (881 Compact IC Pro, Metrohm) equipped with conductivity detectors, respectively. PME sample information is provided in Table S1, while DOC and ion concentrations are in Table S2.

### 2.3 Sample illumination and chemical analysis

Illumination experiments were conducted using light from a 1000 W xenon arc lamp that was passed through optical filters to simulate tropospheric sunlight (Kaur and Anastasio, 2017). 1.0 mL of extract at pH 4.2 was spiked with a photooxidant probe and illuminated in a silicone-plugged GE 021 quartz tube (5 mm inner diameter, 1.0 mL volume) at 20 °C. Dark control samples were wrapped in aluminum foil and kept in the same photoreactor chamber. During illumination, aliquots were removed from the

illuminated and dark tubes periodically to measure probe concentrations with high-performance liquid chromatography (HPLC, Shimadzu LC-20AB pump, Thermo Scientific Accucore XL C18 column (50 × 3 mm, 4 µm bead), and Shimadzu-M20A UV-Vis detector). The photon flux in an identical quartz tube was determined on each experiment day by measuring the photolysis rate constant of a 10 µM 2-nitrobenzaldehyde (2NB) solution (Galbavy et al., 2010).



### 2.4 Photooxidant measurements

Photooxidant methods are detailed in past papers (Anastasio and McGregor, 2001; Kaur and Anastasio, 2017; Ma et al., 2023a) and are only briefly described here. The uncertainty on an individual oxidant concentration is 1 standard error, determined by propagating the errors of the individual parameters required to calculate the concentration. Uncertainties on average values are 1 standard deviation, calculated from the spread of the individual values.

### 2.4.1 Hydroxyl radical ($^\bullet$OH)

$^\bullet$OH concentration was quantified using 10 μM benzoic acid (BA) as a probe and simultaneously monitoring the rates of probe decay and product ($p$-hydroxybenzoic acid, $p$-HBA) formation. For dilute samples (DOC < 15 mg C L$^{-1}$), 2 μM BA was used in order not to perturb the natural $^\bullet$OH sink in PME. Aliquots were taken during illumination to measure BA and $p$-HBA concentrations. From the BA probe loss, a linear regression of $\ln([BA]_t/[BA]_0)$ versus illumination time ($t$) was fitted, where $[BA]_0$ is the concentration at time zero. The negative value of the regression slope is the BA pseudo-first order decay rate constant ($k'_{BA}$).

The $^\bullet$OH concentration was then determined using:

$$[^\bullet OH]_{exp} = \left[\frac{k'_{BA}}{k_{BA+^\bullet OH}}\right] \tag{1}$$

where $k_{BA+\cdot OH}$ is the second-order rate constant of BA reacting with $^\bullet$OH at pH 4.2 ($5.1 \times 10^9$ M$^{-1}$ s$^{-1}$) (Ashton et al., 1995; Wander et al., 1968). Next, $[^\bullet OH]_{exp}$ was normalized to sunlight conditions at midday on the winter solstice at Davis (solar zenith = 62°; $j_{2NB,win}$ = 0.0070 s$^{-1}$) (Galbavy et al., 2010) and corrected for internal light screening due to absorption by chromophores in PME:


$$[^\bullet OH]_{win} = \left[\frac{[^\bullet OH]_{exp}}{S_\lambda \times j_{2NB,exp}}\right] \times j_{2NB,win} \tag{2}$$

where $S_\lambda$ is the internal light screening factor in an individual sample (Table S1), and $j_{2NB,exp}$ is the photolysis rate constant of 2NB measured on the experiment day.

We also determined the $^\bullet$OH concentration in each sample from $p$-HBA formation. The initial formation rate of $p$-HBA was

determined from the regression between $p$-HBA concentration and illumination time, either using a linear regression or a three-parameter exponential fit:

$$[p\text{-}HBA]_t = [p\text{-}HBA]_0 + a(1 - e^{-bt}) \tag{3}$$

where $[p\text{-}HBA]_t$ and $[p\text{-}HBA]_0$ are the measured concentrations at illumination times $t$ and zero, respectively, and $a$ and $b$ are regression fit parameters. With this fitting, the initial formation rate of $p$-HBA, $R_p$, is calculated with:


$$R_P = a \times b \tag{4}$$

and then the $^\bullet$OH concentration was calculated using:

$$[^\bullet OH]_{exp} = \frac{R_p}{[BA]_0 \times k_{BA+\bullet OH} \times Y_{p-HBA}} \tag{5}$$

where $Y_{p\text{-HBA}}$ (0.18) is the yield of $p$-HBA from the reaction of BA with $^\bullet$OH (Anastasio and McGregor, 2001). $^\bullet$OH concentrations were normalized by $j_{2NB}$ and light screening factor using Eq.2. In some samples, BA decay and $p$-HBA formation were faster at

the beginning of illumination and then slowed (e.g., Fig. S1), indicating an initially higher $^\bullet$OH concentration compared to later times, as seen previously (Paulson et al., 2019). For each sample we generally used all data points for the regressions of BA and $p$-HBA and then determined the reported [$^\bullet$OH] as the average of the BA and $p$-HBA results (Table S3).



### 2.4.2 Singlet molecular oxygen ($^1O_2^*$)

To determine $^1O_2^*$ concentrations, FFA was used as a probe and deuterium oxide ($D_2O$) was used as a diagnostic tool (Anastasio and McGregor, 2001) because $^1O_2^*$ decays more rapidly in $H_2O$ than $D_2O$. Therefore, the difference of FFA decay rates in $H_2O$ and $D_2O$ is attributed to $^1O_2^*$ (instead of other oxidants). For each sample, 1.0 mL of PME was divided into two 0.5 mL aliquots, with one diluted with 0.5 mL $H_2O$ and the other 0.5 mL $D_2O$. 10 µM FFA was spiked into both solutions and pseudo-first order rate constants of FFA loss during illumination were determined ($k_{exp,H2O}$ and $k_{exp,D2O}$). The difference between the FFA first-order rate constants was used to calculate the steady-state $^1O_2^*$ concentration (Anastasio and McGregor, 2001). This experimental $^1O_2^*$ concentration was normalized by photon flux and light screening factors of PME using an analog of Eq. 2 to determine $^1O_2^*$ winter-solstice values (Table S4).

### 2.4.3 Oxidizing triplet excited states of organic matter ($^3C^*$)

Oxidizing triplets were measured with two probes, syringol (SYR) and (phenylthio)acetic acid (PTA). SYR reacts rapidly with all oxidizing triplets, but its decay by $^3C^*$ can be inhibited by high concentrations of dissolved organic matter (DOM) (Ma et al., 2023a, 2023b; Maizel and Remucal, 2017; McCabe and Arnold, 2017). In contrast, PTA is more resistant to this inhibition, but it can only capture strongly oxidizing triplets (Ma et al., 2023b). To determine $^3C^*$ concentrations, two 1.0 ml aliquots of PME were spiked with 10 µM of either SYR or PTA, and then illuminated to determine the pseudo-first order rate constant for loss of each probe ($k'_{P,exp}$). We then removed the contributions of direct photodegradation, $^\bullet OH$, and $^1O_2^*$ to triplet probe decay (Ma et al., 2023a). Since $^3C^*$ is a complex mixture of triplets with a wide range of reactivities, there is no exact value for the second-order rate constant of $^3C^*$ in PME reacting with probes. Our past work indicated that $^3C^*$ in Davis winter PM have a similar average reactivity to the triplet state of DMB (Kaur and Anastasio, 2018; Kaur et al., 2019), which is a component of BB BrC (Fleming et al., 2020; Schauer et al., 2001). However, it is possible that this model compound is more reactive than natural oxidizing triplets, which would lead to an underestimate of $^3C^*$ (Ma et al., 2023b). We quantified the inhibition effect of DOM on the decay of SYR and PTA by measuring inhibition factors of each probe ($IF_{P,corr}$) in each sample, and used them to correct $^3C^*$ concentrations (Canonica and Laubscher, 2008; Ma et al., 2023b; McCabe and Arnold, 2017; Wenk et al., 2011). Details about determining inhibition factors and correcting $^3C^*$ concentrations are provided in Supplemental Information Section S1. $^3C^*$ concentrations in PME during each experiment were calculated with:

$$[^3C^*]_{P,exp} = \frac{k'_{P,3C*}}{k_{P+3DMB*} \times IF_{P,corr}} \tag{6}$$

where $k_{P+3DMB*}$ is the second-order rate constant of probe with $^3DMB^*$ (Table S5). These values were converted to $^3C^*$ concentrations expected on midday of the winter solstice in Davis (after correction for internal light screening) using an equation analogous to Eq. 2; these are the concentrations reported in the main text. Details of $^3C^*$ measurements by SYR and PTA are in Tables S7 and S8, respectively.

### 2.4.4 Extrapolating extract results to aerosol liquid water conditions

Photooxidant concentrations in PM extracts represent dilute conditions similar to cloud/fog waters, while our goal is to estimate photooxidant concentrations in aerosol liquid water, which is orders of magnitude more concentrated. To predict photooxidant concentrations in ALW, we quantified photooxidant kinetics (i.e., oxidant formation rates and loss rate constants) for each sample type as a function of particle mass concentration and then extrapolated to ALW conditions (Kaur et al., 2019; Ma et al., 2023a). Details about the extrapolations are provided in Section S4.



## 3. Results and Discussion

### 3.1 General extract characteristics

To investigate the seasonal variation of photooxidant formation, we studied 18 PM$_{2.5}$ samples across a year of sampling. Samples were from all seasons, but there was only one Spring sample because of COVID restrictions from March through June of 2020 (Fig. 1 and Table S1). Most particle samples were collected for 24 h, while four of the winter samples were collected for seven days to obtain more particle mass. Winters were marked by residential wood burning and high relative humidities, while the summer samples represented both periods influenced by fresh and aged biomass burning (from wildfires) and clean conditions. From August to October 2020, Davis periodically experienced severe air pollution caused by wildfires in California and Oregon. Section S2 of the supplement provides satellite images with fire points detected by the NASA Visible Infrared Imaging Radiometer Suite (VIIRS) and 24-h back trajectories estimated by the Hybrid Single Particle Lagrangian Integrated Trajectory (HYSPLIT) on the day of sampling for wildfire periods (Rolph et al., 2017; Stein et al., 2015). Based on the satellite images and back trajectories, smoke plumes were transported from their sources to Davis in as short as 1~2 h or as long as 12~24 h or more.

Figure 1 shows the average PM$_{2.5}$ concentration during each extract sampling period. We categorized the 18 samples into four groups based on sampling date and positive matrix factorization (PMF) results obtained using UV/Vis absorption spectra and aerosol mass spectrometer chemical characterization (Jiang et al., 2023). The first group is termed Winter & Spring samples (Win-Spr), which were collected from November 2019 to March 2020 and have an average PM$_{2.5}$ concentration of 9.9 ($\pm$ 1.5) µg m$^{-3}$ (Table S1). Three samples collected in July, August, and October without wildfire influence are classified as Summer & Fall samples (Sum-Fall), with an average PM$_{2.5}$ of 7.4 ($\pm$ 0.4) µg m$^{-3}$. The seven wildfire-influenced samples collected from August to October are classified as fresh biomass burning (FBB) or aged biomass burning (ABB), with average PM$_{2.5}$ values of 55 ($\pm$ 10) and 24 ($\pm$ 8) µg m$^{-3}$, respectively. The PMF results indicate that FBB samples are dominated by biomass-burning organic aerosol factors characterized by elevated levels of levoglucosan (m/z 60) signature ions in the AMS mass spectra (Alfarra et al., 2007). ABB samples were also collected during the wildfire-influenced period, but they are dominated by an oxidized organic aerosol factor with high O/C ratio and little levoglucosan (Jiang et al., 2023).

Our PM extracts are much more dilute than aerosol liquid water in the ambient atmosphere, a result of physical limitations on the amount of water we need to extract and study particle photochemistry. Particle mass/liquid water mass ratios of our extracts were in the range $(0.7 - 4.1) \times 10^{-4}$ µg PM/µg H$_2$O for one-day samples (Fig. S9) and correlated well with the ambient PM$_{2.5}$ concentrations (Table S1). The seven-day winter samples had higher particle mass/water mass ratios, up to $9.1 \times 10^{-4}$ µg PM/µg H$_2$O. Based on the PM mass concentrations, our particle extracts are similar to dilute atmospheric waters such as cloud and fog drops (10$^{-5}$ - 10$^{-3}$ µg PM/µg H$_2$O), instead of concentrated particle liquid water (roughly 1 µg PM/µg H$_2$O) (Nguyen et al., 2016; Seinfeld and Pandis, 2016).




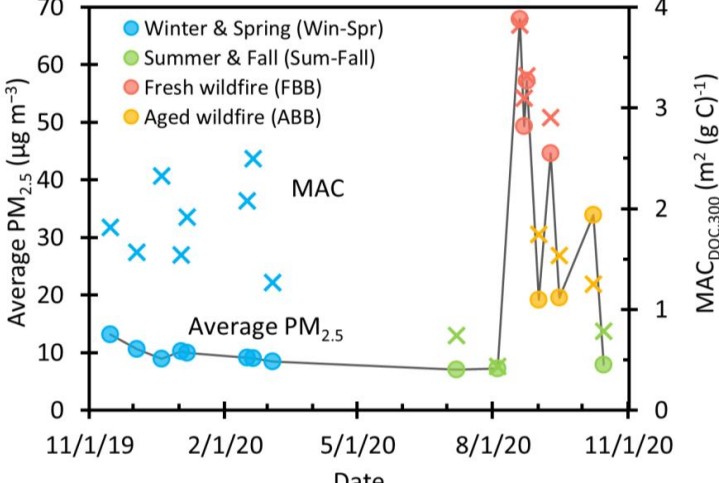

**Figure 1. Average PM$_{2.5}$ concentrations (circles) during each sampling period and DOC-normalized mass absorption coefficients at 300 nm (×) in particle extracts for Winter & Spring samples (blue), Summer & Fall samples (green), Fresh biomass burning (red), and Aged biomass burning (yellow).**

Dissolved organic carbon concentrations in the extracts range from 5 to 192 mg C L$^{-1}$ (Table S1). The ratio of organic carbon (OC) mass to total extracted PM mass is high in the wildfire samples, with average values of 31 ($\pm$ 6) % and 26 ($\pm$ 6) % for FBB and ABB, respectively. These fractions are lower than values for BB particles in other studies (43-59%) (Schauer et al., 2001; Vicente et al., 2013), probably because we used water as the extraction solvent, thereby missing water insoluble organics. The OC/PM fractions for Win-Spr and Sum-Fall samples are similar to each other, with values of 16 ($\pm$ 5) % and 11 ($\pm$ 3) %, respectively. Win-Spr PMEs have high concentrations of nitrate (NO$_3^-$), 84-3300 $\mu$M (Table S2), which contributed up to 33 % of PM mass. PMEs in the other three groups have nitrate concentrations from 25 to 300 $\mu$M, which are less than 10 % of PM mass. Win-Spr samples also have the highest ammonium concentrations, 168-4900 $\mu$M, followed by wildfire-influenced samples (46-803 $\mu$M), and Sum-Fall samples (< 100 $\mu$M). Potassium, a marker of biomass burning (Silva et al., 1999), has its highest concentrations in winter and wildfire samples with a range of 62-220 $\mu$M. The Sum-Fall samples have the highest fraction of sodium with an average of 11%, suggesting the influence of sea salt (Parworth et al., 2017). We employed three field blanks in this study at the beginning, middle, and end of the sampling campaign. In field blanks, ions and DOC concentrations are less than 10% of their concentrations in most PME samples, though FB1 was contaminated by the filling solution of a pH electrode, resulting in extremely high chloride concentrations (Table S2).

## 3.2 Light absorption in particle extracts

DOC-normalized mass absorption coefficients at 300 nm (MAC$_{DOC,300}$) are shown in Figure 1. For wildfire samples, MAC is correlated with the PM$_{2.5}$ concentration, which probably reflects the dominant influence of BB emissions on both PM levels and light absorbance since FBB has the highest MAC among sample types, with an average of 3.3 ($\pm$ 0.4) (g C)$^{-1}$. This is expected because fresh biomass burning organic aerosols (BBOA) contain abundant amounts of highly light-absorbing products, including substituted aromatics with high unsaturation and nitroaromatics (Budisulistiorini et al., 2017; Claeys et al., 2012; Fleming et al., 2020; Hettiyadura et al., 2021; Lin et al., 2016, 2017). The average MAC for FBB at 365 nm is 1.2 ($\pm$ 0.4) m$^2$ (g C)$^{-1}$, similar to past values determined in water extracts of biomass burning particles (0.9 – 1.4 m$^2$ (g C)$^{-1}$) (Du et al., 2014; Fan et al., 2018; Park and Yu, 2016). At 300 nm, the average MAC of ABB is 1.5 ($\pm$ 0.3) m$^2$ (g C)$^{-1}$, half the value of FBB, likely because of





photobleaching of brown carbon during aging (Hems and Abbatt, 2018; Hems et al., 2021; Laskin et al., 2015; Wong et al., 2017;

Zhao et al., 2015). Win-Spr has an average $MAC_{DOC,300}$ (1.9 (± 0.4) m$^2$ (g C)$^{-1}$) that is three times higher than that of Sum-Fall (0.65 (± 0.19) m$^2$ (g C)$^{-1}$), though they have similar $PM_{2.5}$ concentrations. This indicates that winter wood combustion can significantly enhance light absorption by particles. Our winter MAC value is similar to the average value (2.2 (± 0.7) m$^2$ (g C)$^{-1}$) determined in previous water extracts of Davis winter particles (Kaur et al., 2019).

We also calculated the average $MAC_{DOC}$ for each sample type in the wavelength range of 300-600 nm, as shown in Figure 2. Fresh wildfire samples have the highest MAC values across the wavelength range and the lowest absorption Ångström exponent (AAE, 300 – 450 nm), which is 7.3 (± 0.2). ABB shows slightly lower MAC values than Win-Spr. This might be explained by faster rates of aging and photobleaching during summer as well as higher amounts of less absorbing SOA. AAE values of ABB and Win-Spr are similar, 7.7 (± 0.3) and 7.9 (± 0.3), respectively, and are comparable to previously reported values of water-soluble organic

carbon from biomass burning (Du et al., 2014; Hecobian et al., 2010; Lin et al., 2017). Sum-Fall has the lowest MAC but the highest AAE (9.1 (±0.5)).

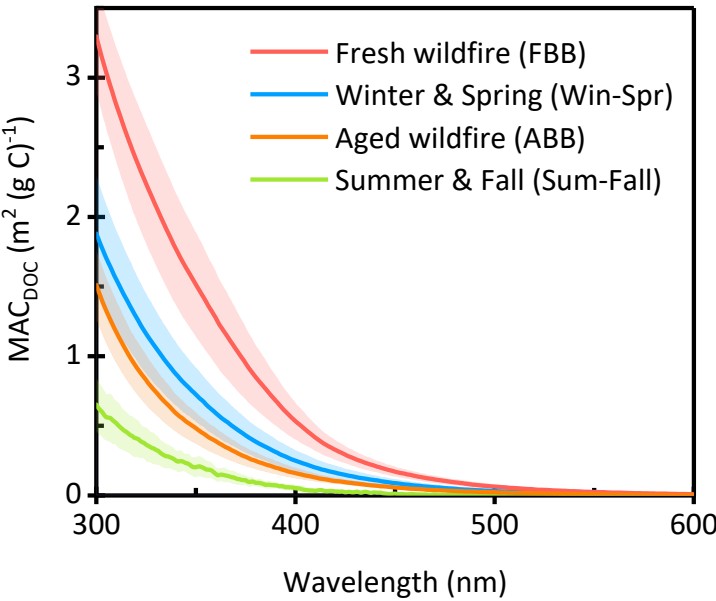

**Figure 2. Average DOC-normalized mass absorption coefficients for Fresh wildfire samples (red), Winter & Spring samples (blue), Aged wildfire samples (orange), and Summer & Fall samples (green). Each shaded area represents ±1 standard deviation.**


An optical property frequently used to characterize surface water DOM is $E_2/E_3$, which is the ratio of absorbance at 250 nm to that at 365 nm. In surface waters, this ratio is an indicator of the molecular weight of dissolved organic matter, with low $E_2/E_3$ representing high molecular-weight DOM (Ossola et al., 2021). $E_2/E_3$ in our PMEs ranges from 4.2 to 17 and is related to MAC values: as shown in Fig. 3, MAC decreases with increasing $E_2/E_3$, i.e., absorbance decreases as DOM molecular weight decreases.

FBB has the lowest average $E_2/E_3$ (5.8 (± 1.5)) of our sample types, including ABB (12.5 (± 2.3)), which suggests that organic molecules in fresh BB are fragmented during aging. This is consistent with the observation that high-molecular weight compounds are less abundant in aged BBOA (Farley et al., 2022). Therefore, $E_2/E_3$ may be an easy and effective indicator to differentiate fresh





and aged samples. $E_2/E_3$ ratios for the Win-Spr samples are intermediate between the summer fresh and aged BB samples, again suggesting these biomass-burning influenced winter samples are less aged than ABB.

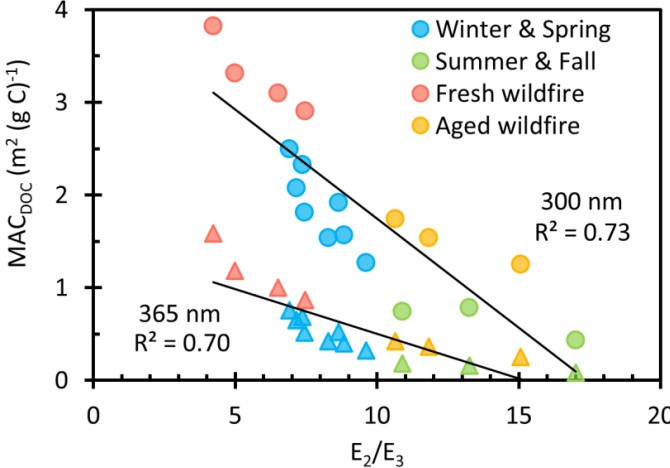


**Figure 3. Mass absorption coefficients of dissolved organic carbon at 300 nm (circles) and 365 nm (triangles) as a function of $E_2/E_3$ for each sample type. Solid lines represent linear regressions.**

Since the light absorption of methanol extracts of particles are usually greater than those of water extracts (Liu et al., 2013; Zhang et al., 2013), we also examined the absorbance of a FBB filter extracted with different solvents (water, methanol, and hexane). As

shown in Fig. S10, the absorbance of the methanol (MeOH) extract is more than twice as high as the water extract, and five times higher than the hexane extract, indicating this FBB contains a high fraction of water-insoluble brown carbon. We also did a sequential extraction with this FBB sample and with a Win-Spr sample, with 1st, 2nd, and 3rd extraction solvents of water, methanol, and hexane, respectively. The UV-Vis spectra and PM mass extracted for each solvent extraction are shown in Fig. S11. For the Win-Spr and FBB samples, the PM mass recovered by the second extraction (in methanol) are only 20% and 56% of the mass by

the first extraction (in water), respectively, but the MeOH extract absorbance at 365 nm is similar or even greater than the water extract. This is consistent with a previous study of sequential extraction with US western wildfire samples (Zeng et al., 2022), which found that water-insoluble brown carbon (e.g., polycyclic aromatic hydrocarbons) is highly light-absorbing, despite accounting for little of the PM mass. The high light absorption in methanol extracts suggests that the water-insoluble chromophores have high potential to produce photooxidants. However, since the oxidant probes we use were developed for aqueous, and not

organic, solutions we did not study photooxidant generation in methanol or hexane extracts.

### 3.3 Photooxidant concentrations

### 3.3.1 Normalization by sample duration

While most of our PM samples were collected for 1 day, we also collected four samples for 7 days, which resulted in extracts that were more concentrated and that had higher oxidant concentrations. To properly compare these longer samples with the rest, we

normalized photooxidant concentrations in the 7-day samples to what would be expected for a 24-h sample. For $^1O_2^*$ and $^3C^*$, the production rate is proportional to the brown carbon mass (Faust and Allen, 1992; Kaur et al., 2019) and so we normalized their concentrations by dividing by the duration of sampling (i.e., number of sampling days). The case for hydroxyl radical is more complicated, since past work has found that the $^\bullet$OH concentration can be independent of extract concentration (Arakaki et al.,





2013; Kaur et al., 2019), but unnormalized $^\bullet$OH concentrations in our 7-day samples are clearly higher than in the adjacent 24-h
samples (Fig. S12). If we normalize $^\bullet$OH using the same method as for $^1O_2^*$ and $^3C^*$ (i.e., by the duration of sampling), the resulting
$^\bullet$OH concentrations are lower than the adjacent 24 h samples (Fig. S12). To obtain more reasonable estimates for [$^\bullet$OH] in the 7-
day samples, we fitted the plot of $^\bullet$OH concentration versus particle mass/water mass ratio for Win-Spr samples with a linear
regression (Fig. S13), and then used the regression to estimate $^\bullet$OH concentrations in the 7-day samples using the time-normalized
particle mass/water mass ratio values (i.e., measured particle mass/water ratio divided by 7).

### 3.3.2 Hydroxyl radical ($^\bullet$OH)

As shown in Fig. 4a, normalized $^\bullet$OH concentrations have a range of $(0.2\text{-}3.2) \times 10^{-15}$ M. The values are similar to those in
illuminated particle extracts from Davis and Colorado (Kaur et al., 2019; Leresche et al., 2021), but much higher than those in
illuminated extracts of lab SOA and $PM_{10}$ from Switzerland $((2.2\text{-}4.9)\times10^{-17}$ M) that had low DOC (5 mg C L$^{-1}$) (Manfrin et al.,
2019). Among our four sample types, fresh biomass burning samples have the highest average [$^\bullet$OH], 2.5 ($\pm$ 0.3) $\times10^{-15}$ M, while
aged BB particles have a similar average concentration that is statistically indistinguishable, 1.7 ($\pm$ 1.4) $\times10^{-15}$ M. This is parallel
to a previous finding that BBOA, compared to other types of organic aerosols, has the highest oxidative potential as measured by
the DTT assay and this potential decreases with simulated atmospheric aging (Verma et al., 2015; Wong et al., 2019). Win-Spr has
a similar average [$^\bullet$OH], 1.5 ($\pm$ 0.3) $\times10^{-15}$ M, while Sum-Fall is the lowest at 0.4 ($\pm$ 0.3) $\times10^{-15}$ M. Our winter values are roughly
three to four times higher than average values in previous Davis winter particle extracts and fog waters (0.51 ($\pm$ 0.24) $\times 10^{-15}$ M
and 0.42 ($\pm$ 0.07) $\times 10^{-15}$ M, respectively) (Kaur and Anastasio, 2017; Kaur et al., 2019). While nitrate and nitrite can be important
sources of $^\bullet$OH in atmospheric waters (Anastasio and McGregor, 2001; Kaur and Anastasio, 2017; Kaur et al., 2019; Leresche et
al., 2021), these species account for less than 10% of $^\bullet$OH in most of our current samples (Table S3). In our kinetic experiments,
in 6 of our 18 samples (5 winter samples and 1 wildfire sample) BA decayed faster at the beginning of irradiation but was slower
at later times, with a rate difference up to a factor of 3.4 (Fig. S1). This indicates [$^\bullet$OH] in some samples is higher during the initial
stage of irradiation, possibly because a portion of the compounds that produce $^\bullet$OH are labile and undergo rapid decomposition.
A similar effect was seen in biomass burning aerosols from Fresno CA, where a burst of $^\bullet$OH was observed within the first few
minutes of irradiation and was hypothesized to be due to the decomposition of peroxides through photo-Fenton reactions (Paulson
et al., 2019).



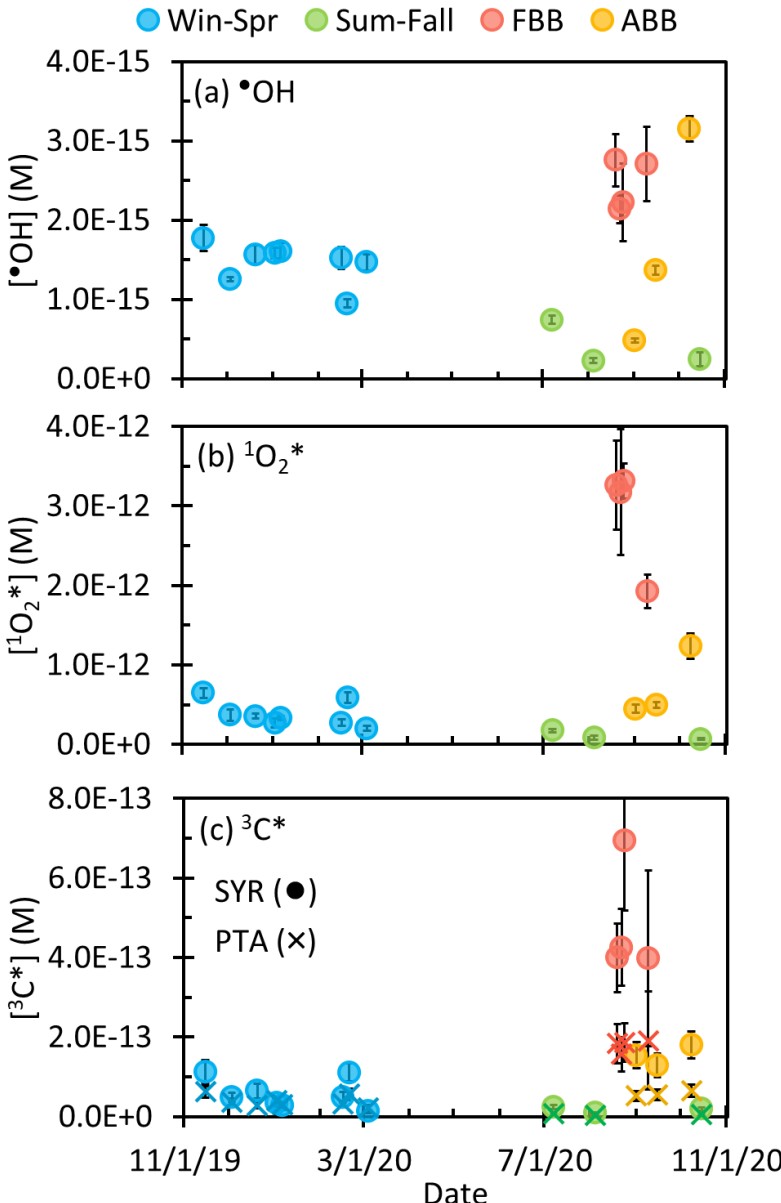

**Figure 4. Steady-state concentrations of (a) hydroxyl radical, (b) singlet molecular oxygen, and (c) oxidizing triplet excited states of organics determined by syringol (circles) and (phenylthio)acetic acid (crosses) in particle extracts. Concentrations are all normalized by sampling duration and to midday winter solstice sunlight in Davis to highlight seasonal differences in particle reactivity; the equivalent plots with concentrations calculated for the midday sunlight of each sample collection period is shown in Figure S14.**

Figure. 5a shows $^\bullet$OH concentration as a function of dissolved organic carbon for the four sample types. For comparison, we also include data from Kaur et al. (2019), who measured photooxidant concentrations in Davis winter particle extracts. Though samples in Kaur et al. (2019) have similar values of DOC as our 24-h Win-Spr samples, their [$^\bullet$OH] is 5 times lower and independent of DOC. While $^\bullet$OH appears to increase with DOC (Fig. 5a), the data is noisy and the linear correlation is weak ($R^2 = 0.40$). A previous study on Minnesota surface waters observed a logarithmic relationship between [$^\bullet$OH] and absorbance coefficient at 440 nm (Chen et al., 2020), which in turn was correlated to DOC. They speculated this is because the dominant $^\bullet$OH sink changes from



bicarbonate/carbonate to DOC with increasing DOC levels, but bicarbonate/carbonate are negligible sinks in our extracts since they are acidic (pH 4.2). [•OH] in FBB is independent of DOC, but the three ABB samples show •OH increasing with DOC. We also found that [•OH] increases with DOC in a dilution series of summer wildfire PM and hypothesized that •OH production is a bimolecular reaction (primarily Fe(II) + HOOH) that increases as the square of PM mass concentration (Ma et al., 2023a). This might also explain our current ABB results.

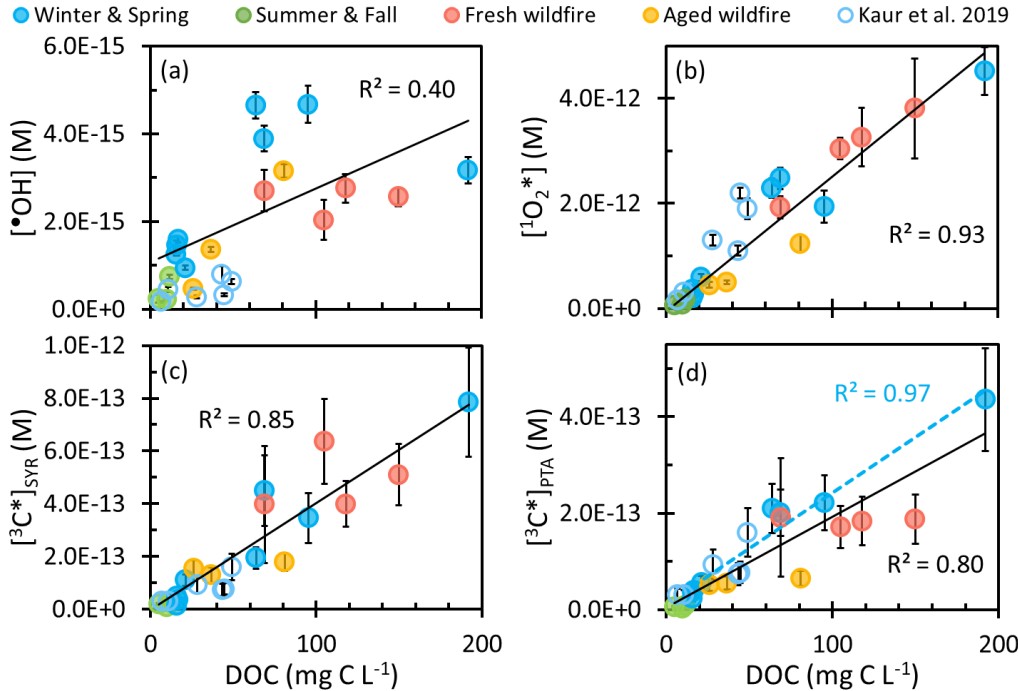

**Figure 5. Steady-state concentrations of (a) hydroxyl radical, (b) singlet molecular oxygen, and oxidizing triplet excited states of organic matter determined by (c) syringol and (d) (phenylthio)acetic acid as a function of dissolved organic matter for each sample type (solid circles). Previous measurements made in Davis winter particle extracts are in open circles (Kaur et al., 2019). Solid black lines are linear regressions between oxidant concentrations in this work and DOC. The blue dashed line in panel (d) is the linear regression of the Win-Spr samples. Error bars represent standard error propagated from linear regression and error in rate constants. Oxidant concentration values are not normalized by the sampling duration.**

### 3.3.2 Singlet molecular oxygen ($^1O_2^*$)

Winter-solstice-sunlight normalized $^1O_2^*$ has a concentration range of $(0.7-32) \times 10^{-13}$ M (Fig. 4b) and correlates well with ambient $PM_{2.5}$ concentration (Fig. S15). These concentrations are similar to the wide range of previously reported values in particle extracts, $(0.6-22) \times 10^{-13}$ M (Bogler et al., 2022; Kaur et al., 2019; Leresche et al., 2021), but are roughly 100 times higher than concentrations in illuminated extracts of biogenic and anthropogenic SOA, $(0.8-45) \times 10^{-15}$ M (Manfrin et al., 2019). Our higher $^1O_2^*$ concentrations are only partially explained by our 1 – 40 times higher DOC concentrations; the remaining difference is likely due to greater light absorption by our samples. Our values are also similar to $[^1O_2^*]$ in solutions of dissolved soot illuminated with simulated sunlight, $(0.6 – 65) \times 10^{-13}$ M (Li et al., 2019), even though their samples absorbed very little light. Among our samples, Fresh BB has the highest average $[^1O_2^*]$, followed by ABB, with values of 29 ($\pm$ 7) $\times 10^{-13}$ M and 7.3 ($\pm$ 0.4) $\times 10^{-13}$ M, respectively. Leresche et al. (2021) found that $[^1O_2^*]$ decreased by a factor of two in particle extracts after sunlight irradiation, which is consistent with our observation that aged particle extracts have lower $[^1O_2^*]$. Win-Spr and Sum-Fall samples have average $[^1O_2^*]$ values of



3.8 (± 1.6) ×$10^{-13}$ M and 1.1 (± 0.6) ×$10^{-13}$ M, respectively. The higher Win-Spr concentrations are probably because of the

influence of biomass burning.

As shown in Fig. 5b, $^1O_2^*$ concentrations linearly increase with DOC ($R^2$ = 0.93), consistent with our understanding that organic

matter is the primary source of $^1O_2^*$ (Bogler et al., 2022; Kaur and Anastasio, 2017; Kaur et al., 2019; Ossola et al., 2021).

Moreover, all four types of samples share the same slope, suggesting the relationship between [$^1O_2^*$] and DOC is independent of

particle type or chemical composition, which is somewhat surprising given the large differences in DOC-normalized light

absorption for the different samples types (Fig. 2). When plotting [$^1O_2^*$] as a function of absorbance at 300 and 365 nm (Figs.

S16b and S17b, respectively), we do observe differences among sample types. In these plots, Win-Spr samples present a steeper

slope (as do samples from Kaur et al. (2019)) compared to wildfire samples, consistent with our previous work (Ma et al., 2023a).

The $^1O_2^*$ concentrations in previous Davis winter particle extracts (Kaur et al., 2019) also follow the linear regression of this work.

While this suggests DOC is a robust descriptor for $^1O_2^*$ concentrations, most of our particle samples were influenced by biomass

burning. Other particle types - such as anthropogenic SOA, biogenic SOA, and emissions from fossil fuel combustion appear to

have different relationships between $^1O_2^*$ and DOC, as suggested by results from Manfrin et al. (2019), Jiang et al. (2023), and

Bogler et al. (2022).

### 3.3.3 Oxidizing triplet excited states of brown carbon ($^3C^*$)

We used two probes − syringol (SYR) and (phenylthio)acetic acid (PTA) − to quantify oxidizing triplet excited states. SYR reacts

rapidly with both strongly and weakly oxidizing triplets, while PTA is only reactive with strongly oxidizing triplets (Ma et al.,

2023b). However, syringol has a disadvantage that its decay by $^3C^*$ can be inhibited by dissolved organic matter, while PTA is

largely resistant to inhibition (Ma et al., 2023b; Maizel and Remucal, 2017; McCabe and Arnold, 2017; Wenk et al., 2011). As

shown in Fig. 4c, winter-solstice-normalized (and inhibition-corrected) $^3C^*$ concentrations have a range of (0.13 − 6.9) ×$10^{-13}$ M

as determined by SYR and (0.03 − 1.9) ×$10^{-13}$ M by PTA. The $^3C^*$ concentration follows $PM_{2.5}$ concentration well, with low values

during non-wildfire periods and very high values during wildfire-influenced periods (Fig. S15). For nearly all samples, [$^3C^*$]$_{SYR}$ is

higher than [$^3C^*$]$_{PTA}$. As seen for $^1O_2^*$, FBB has the highest average [$^3C^*$], 4.8 (± 1.4) ×$10^{-13}$ M from SYR and 1.8 (± 1.6) ×$10^{-13}$

M from PTA, due to the high organic amounts in these samples. Relative to the FBB average, the FBB, ABB, Win-Spr, and Sum-

Fall samples have triplet concentration ratios of 1 : 0.32 : 0.12 : 0.04 as determined by SYR and 1 : 0.32 : 0.21 : 0.03 as determined

by PTA. These ratios are similar to the ratio of average DOC concentrations, which is 1: 0.45: 0.15: 0.08, indicating DOC is the

main driver of $^3C^*$ concentration differences among sample types. This relationship is complicated at high DOC where dissolved

organics can be the dominant triplet sink (up to roughly 60% of the total sink), larger than the contribution from dissolved oxygen.

Figure 5c shows the correlation between [$^3C^*$]$_{SYR}$ and DOC for our samples, along with data from Kaur et al. (2019). [$^3C^*$]$_{SYR}$

linearly increases with DOC ($R^2$ = 0.83) independent of sample type, likely because SYR reacts rapidly with a wide range of

oxidizing triplets (Kaur and Anastasio, 2018). However, Figs. S16c and S17c show some differences between sample types in the

relationship between [$^3C^*$]$_{SYR}$ and absorbance at 300 or 365 nm, with Win-Spr samples having a steeper slope. However, the trend

of FBB samples is hard to discern, in part because of the limited number of samples available (only four). As shown in Fig. 5d,

[$^3C^*$]$_{PTA}$ also linearly increases with DOC, though the correlation is not as good as those for [$^1O_2^*$] or [$^3C^*$]$_{SYR}$. Win-Spr samples

present a slightly higher slope than wildfire samples (FBB and ABB); oddly, [$^3C^*$]$_{PTA}$ is nearly independent of DOC within either

biomass burning group. The steeper slope of [$^3C^*$]$_{PTA}$ with DOC for the Win-Spr samples suggests these samples contain a higher



fraction of highly oxidizing $^3C^*$ than the wildfire samples. This difference in slopes is particularly noticeable in Figs. S16d and S17d, where $[^3C^*]_{PTA}$ is plotted against absorbance at 300 or 365 nm.

Since PTA only captures $^3C^*$ that have high reduction potentials, while SYR reacts rapidly with both strongly and weakly oxidizing triplets, the ratio $[^3C^*]_{PTA}/[^3C^*]_{SYR}$ provides an estimate of the fraction of oxidizing $^3C^*$ that are strong oxidants. As shown in Fig. 6, the ratio $[^3C^*]_{PTA}/[^3C^*]_{SYR}$ ranges from 0.27 ($\pm$ 0.10) to 1.7 ($\pm$ 0.7) with an average value of 0.58 ($\pm$ 0.38), indicating roughly 60% of oxidizing triplets are strong oxidants. The Win-Spr samples have an average ratio of 0.86 ($\pm$ 0.43), significantly higher than the rest of the samples (0.37 $\pm$ 0.07), indicating that they produce a higher fraction of strongly oxidizing $^3C^*$. Precursors for

more oxidizing triplets include quinones, aromatic ketones and aromatic aldehydes, while weakly oxidizing triplet precursors include polycyclic aromatic compounds (McNeill and Canonica, 2016).

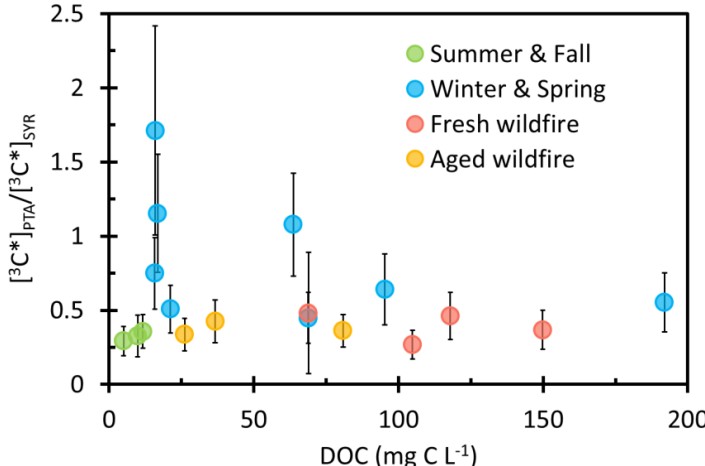

**Figure 6. The ratio of oxidizing triplet excited state concentrations determined by PTA to those determined by SYR as a function of DOC for each sample type.**

We can also gain some insight into extract compositions from the inhibition factors (*IF*) (Section S1) for SYR and PTA in each sample. An *IF* of 1 represents no inhibition of probe decay by the sample, while an *IF* of 0 indicates that the triplet-mediated decay of probe is completely reversed by DOM in the sample (Canonica and Laubscher, 2008; Ma et al., 2023b). Among our samples, *IF* for SYR (*IF*$_{SYR,corr}$) ranges from 1.2 to 0.21, with an average value of 0.64 ($\pm$ 0.29) (Table S6 and Fig. S18). This indicates that SYR decay by $^3C^*$ in PME can be heavily inhibited, suggesting that our PMEs contain abundant antioxidants such as phenolic

or aniline moieties (Wenk and Canonica, 2012; Wenk et al., 2011). As shown in Fig. S18b, *IF*$_{SYR,corr}$ generally decreases with increasing DOC, consistent with previous surface water studies (Canonica and Laubscher, 2008; McCabe and Arnold, 2017). We fit *IF*$_{SYR,corr}^{-1}$ versus DOC using a linear regression with all samples (Ma et al., 2023b; Wenk et al., 2011) as shown in Fig S18b. The fitted slope is 0.015 L mg C$^{-1}$; the inverse of this slope, 67 ($\pm$ 13) mg C L$^{-1}$, represents the DOC concentration that causes *IF*$_{SYR,corr}$ to equal 0.5. All of the sample groups essentially fit on the same line. The *IF* for PTA (*IF*$_{PTA,corr}$) ranges from 1.5 to 0.6,

with an average value of 1.1 ($\pm$ 0.2), demonstrating its better resistance to inhibition (Fig. S18c). We also measured the inhibition factor of furfuryl alcohol (*IF*$_{FFA}$) as the indicator of the ability of DOM in PME to quench $^3C^*$ (Fig. S18a). *IF*$_{FFA}$ decreases with increasing DOC, ranging from 1.4 (i.e., no quenching of triplets by PME DOM) to 0.5 (i.e., DOM is reducing the triplet concentration to 50 % of its non-quenched value). From the linear fit between *IF*$_{FFA}^{-1}$ and DOC, we obtain a second-order rate



constant of DOM quenching $^3$DMB* (Ma et al., 2023b; Wenk et al., 2011, 2013) of 2.7 ($\pm$ 0.7) $\times 10^7$ L (mol-C) s$^{-1}$. This value is
somewhat lower than rate constants of DOM quenching oxidizing $^3$C* in Davis particle extracts ((5.7 – 12) $\times 10^7$ L (mol C)$^{-1}$ s$^{-1}$)
(Ma et al., 2023a) but in the range of values for DOM quenching $^3$C* in surface waters. (1.3-7.9) $\times 10^7$ L (mol C)$^{-1}$ s$^{-1}$ (Wenk et al.,
2013).

### 3.3.5 Normalization by photon flux

Photooxidant concentrations in Figures 4 and 5 are all normalized to the same actinic flux condition (i.e., solar noon on the winter
solstice in Davis CA, $j_{2NB}$ = 0.007 s$^{-1}$) to highlight seasonal differences in particle reactivity. However, photon fluxes vary
throughout the year, which will affect the rate of photooxidant formation and accompanying concentration. To account for this
effect, we calculated midday $j_{2NB}$ values as a function of date during our sampling campaign, as shown in Fig. S19 and described
in Section S3. The estimated $j_{2NB}$ value at midday of the summer solstice is 0.013 s$^{-1}$, which is nearly twice the value during winter.
Next, we estimated midday $j_{2NB}$ values for each sampling day and normalized photooxidant concentrations to the corresponding
sunlight condition. Figure S14 shows the equivalent plot of Figure 3 after photon flux normalization, which increased oxidant
concentrations by factors ranging from 1.0 to 1.9. The average normalization factors for FBB and Sum-Fall samples are 1.7, while
ABB and Win-Spr have average factors of 1.5 and 1.2, respectively. These $j_{2NB}$ values do not account for optical confinement of
sunlight within particles; recent work suggests that this will enhance in-particle actinic fluxes by approximately a factor of two
(Corral Arroyo et al., 2022), which would cause a proportional increase in oxidant concentrations. At this point we do not have
enough information to understand how seasonal variations in temperature might affect oxidant concentrations, so we have not
attempted to factor this into our analysis.

### 3.4 Quantum yields for photooxidants

### 3.4.1 Hydroxyl radical

To investigate how sample type affects the efficiency of photooxidant formation, we determined apparent quantum yields of
photooxidant formation ($\Phi_{Ox}$), i.e., the fraction of absorbed photons that result in formation of a particular photooxidant:

$$\Phi_{Ox} = \frac{P_{Ox}}{R_{abs}} \qquad (7)$$

where $P_{Ox}$ is the oxidant production rate and $R_{abs}$ is the rate of sunlight absorption by the sample between 300 and 450 nm (Kaur
et al., 2019). We calculate the production rate of $^\bullet$OH, $P_{OH}$, by assuming it is equal to the $^\bullet$OH consumption rate since hydroxyl
radical (and the other photooxidants) are at steady state. Thus, $P_{OH}$ is equal to the product of [$^\bullet$OH] and the first-order rate constant
of $^\bullet$OH loss by natural sinks ($k'_{OH}$). To estimate $k'_{OH}$, we assume that organic matter is the dominant sink for $^\bullet$OH (Kaur et al.,
2019) and that $k'_{OH}$ is the product of DOC concentration and the second-order rate constant of DOC with $^\bullet$OH ($k_{DOC+OH}$). For
$k_{DOC+OH}$, we used the average value measured in Davis winter and summer wildfire particle extracts (Ma et al., 2023a), which is
2.7 ($\pm$ 0.4) $\times 10^8$ L (mol-C)$^{-1}$ s$^{-1}$. This value is slightly lower than that determined by Arakaki et al. (2013) for a broad range of
atmospheric waters (3.8 ($\pm$ 1.9) $\times 10^8$ L (mol-C)$^{-1}$ s$^{-1}$) and the one from Leresche et al. (2021) for Colorado PM extracts (4.9 ($\pm$ 2.3)
$\times 10^8$ L (mol-C)$^{-1}$ s$^{-1}$), but none of these are statistically different. In our samples, the resulting calculated $k'_{OH}$ is in the range (0.11
– 4.3) $\times 10^6$ s$^{-1}$ (Table S3), yielding $P_{OH}$ in the range of (0.04-14) $\times 10^{-9}$ M s$^{-1}$, similar to past measured and modeled values for
fog/cloud waters and particle extracts (Arakaki et al., 2013; Leresche et al., 2021; Tilgner and Herrmann, 2018).




Our calculated apparent quantum yields of •OH are shown in Fig. 7a, along with past Davis winter PME samples from Kaur et al.
(2019). $\Phi_{OH}$ ranges from 0.01 % to 0.10 % in our samples, which are generally higher than values from Kaur et al. (2019) and from
$PM_{10}$ and lab SOA water extracts (Manfrin et al., 2019). As expected, $\Phi_{OH}$ appears independent of DOC. Average •OH quantum
yields for Win-Spr, Sum-Fall, FBB, and ABB are 0.044 (± 0.022) %, 0.028 (± 0.010) %, 0.021 (±0.005) %, and 0.049 (±0.050) %,
respectively. These averages are not statistically different ($p > 0.05$).

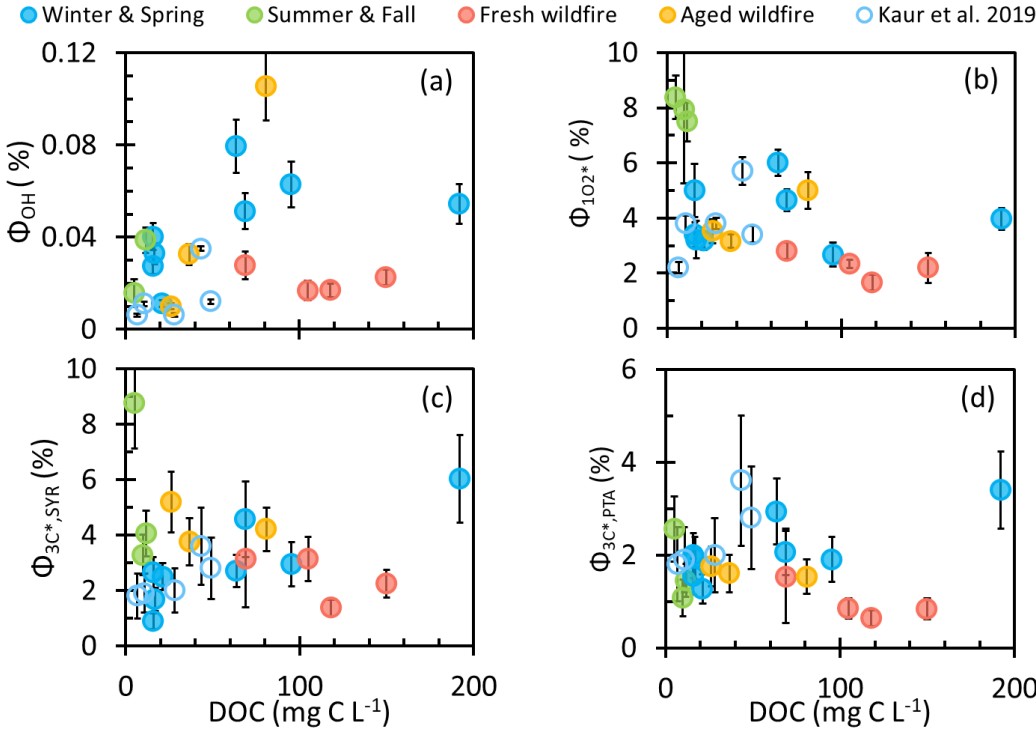

**Figure 7. Apparent quantum yields of (a) hydroxyl radical, (b) singlet molecular oxygen, and oxidizing triplets determined by (c) syringol
and (d) (phenylthio)acetic acid as a function of dissolved organic matter for each sample type (solid circles). Previous measurements
made in Davis winter particle extracts are in open circles (Kaur et al., 2019).**

### 3.4.2 Singlet molecular oxygen

To calculate the apparent quantum yields of $^1O_2^*$ ($\Phi_{1O2^*}$), we assume that $H_2O$ is the dominant sink for $^1O_2^*$ in our PM extracts.
This is a reasonable assumption since the first-order rate constants for $^1O_2^*$ loss via DOC are $(0.04 - 2) \times 10^3$ s$^{-1}$ in our samples
(based on an estimated $^1O_2^* + DOC$ rate constant of $1 \times 10^5$ L (mol-C)$^{-1}$ s$^{-1}$ (Ma et al., 2023a)), while the rate constant for $^1O_2^*$
loss by water is $2.2 \times 10^5$ s$^{-1}$ (Bilski et al., 1997). Therefore, we calculated the production rate of $^1O_2^*$ ($P_{1O2^*}$) by multiplying the
rate of $^1O_2^*$ loss by water ($k'_{H2O}$) by $[^1O_2^*]$. As shown in Fig. 7b, $\Phi_{1O2^*}$ ranges from 1.7% to 8.4%, comparable to values from
Kaur et al. (2019), which are shown as open circles in the figure, as well as from SOA and ambient particle extracts in other studies
(0.1 – 4.5 %) (Bogler et al., 2022; Leresche et al., 2021; Manfrin et al., 2019). But our $\Phi_{1O2^*}$ values are significantly lower than
those in dissolved soot extracts (33%) (Li et al., 2019) under 377 nm irradiation; we do not expect significant black carbon in our
extracts since they were filtered. Sum-Fall has the highest average $\Phi_{1O2^*}$, 7.9 (± 0.4) %, which is significantly different from the
others, while Win-Spr and ABB have similar average values, 4.0 (± 1.1) and 3.9 (± 1.0), respectively, and FBB shows the lowest
average $\Phi_{1O2^*}$ of 2.2 (± 0.5) %. The higher quantum yield for aged biomass burning PM compared to fresh BB PM is broadly
consistent with the enhancement in $\Phi_{1O2^*}$ resulting from ozonation of surface water DOM (Leresche et al., 2019). The difference



among sample types is more pronounced when $\Phi_{1O2*}$ is plotted as a function of MAC. As shown in Figs. S20b and S21b, $\Phi_{1O2*}$ decreases with absorbance at 300 or 365 nm, indicating that less light-absorbing brown carbon (e.g., Sum-Fall) more efficiently produces $^1O_2*$ compared to high-MAC samples (e.g., FBB). In surface waters, $\Phi_{1O2*}$ is positively correlated with $E_2/E_3$, i.e., the

$^1O_2*$ quantum yield increases for DOM with lower average molecular weight molecules (Berg et al., 2019; Ossola et al., 2021). We find a similar linear relationship in our samples, with an $R^2$ of 0.54 (Fig. S22). The fresh BB extract has low $E_2/E_3$ (and low $\Phi_{1O2*}$), suggesting that it contains more high-molecular-weight compounds that absorb significant amounts of light but inefficiently produce $^1O_2*$. It has been suggested that DOM with a high lignin content (as expected for BB PM) can have a high degree of charge transfer interactions, which results in low $\Phi_{1O2*}$ (Ossola et al., 2021). Despite the relatively inefficient production of singlet

oxygen by the fresh BB extracts, these samples have some of the highest $^1O_2*$ concentrations (Fig. 4), a result of their very strong light absorption (Fig. 2).

### 3.4.3 Oxidizing triplet excited states

To calculate the production rate of $^3C*$, we first need to estimate the $^3C*$ sink, which is dominated by dissolved oxygen at low DOC but by organic matter as DOC increases. We estimated average second-order rate constants for DOC reacting with and

physically quenching $^3C*$ ($k_{rxn+Q,3C*}$) in our samples by fitting $[^3C*]$ as a function of DOC with a hyperbolic regression (Fig. S23). Values of $k_{rxn+Q,3C*}$, calculated from one of the regression fitting parameters (Kaur et al., 2019), are 7.2 ($\pm$ 2.2) $\times 10^7$ L (mol-C)$^{-1}$ s$^{-1}$ for $^3C*$ determined by SYR and 7.4 ($\pm$ 2.5) $\times 10^7$ L (mol-C)$^{-1}$ s$^{-1}$ for $^3C*$ by PTA. Since the production rate of $^3C*$ ($P_{3C*}$) is equal to its loss rate, we calculate the former with:

$$P_{3C*} = (k_{rxn+Q,3C*}[DOC] + k_{3C*+O2}[O_2]) \times [^3C*] \qquad (8)$$

where $k_{3C*+O2}$ is the second-order rate constant of dissolved oxygen reacting with $^3C*$ (2.8 $\times 10^9$ M$^{-1}$ s$^{-1}$) (Kaur et al., 2019) and $[O_2]$ is the dissolved oxygen concentration, 280 µM at 20 °C for an air-saturated solution (U.S. Geological Survey, 2020). The apparent quantum yield of $^3C*$ is then calculated using $P_{3C*}$ divided by the rate of light absorption (Eq. 7).

Figures 7c and 7d show quantum yields of $^3C*$ determined by SYR ($\Phi_{3C*,SYR}$) and PTA ($\Phi_{3C*,PTA}$). $\Phi_{3C*,SYR}$ has a range of (0.9-8.8)

% and an average value of 3.5 ($\pm$ 1.8) %. Our values are similar to $\Phi_{3C*}$ in past Davis winter PM extracts (as shown by the open circles in the figures), as well as fog waters and surface waters, which are in the range (0.3-14) % (Kaur and Anastasio, 2018; McCabe and Arnold, 2018). We do not observe significant differences in $\Phi_{3C*,SYR}$ among sample types (Fig. S24), consistent with the similarities among sample types in the relationship of $[^3C*]_{SYR}$ versus DOC (Fig. 4). $\Phi_{3C*,PTA}$ has a range of (0.6-3.4) %, with an average value of 1.7 ($\pm$ 0.7) %, half of the average $\Phi_{3C*,SYR}$. Win-Spr has the highest average $\Phi_{3C*,PTA}$, 2.1 ($\pm$ 0.7) %, while FBB

has the lowest, 0.96 ($\pm$ 0.39) %, but they are not statistically different. Though $^3C*$ is the precursor of $^1O_2*$, $\Phi_{3C*}$ does not correlate well with MAC, unlike $\Phi_{1O2*}$ (Figs. S20 and S21), probably because we are measuring only the oxidizing portion of the triplet pool. In surface waters, $\Phi_{3C*}$ often increases with $E_2/E_3$, similar to $\Phi_{1O2*}$ (Berg et al., 2019; Maizel and Remucal, 2017; McCabe and Arnold, 2017), but we do not see this triplet behavior in our samples (Fig. S25) even though we do for $^1O_2*$ (Fig. S22).

We next use our quantum yields to estimate the fraction of the total triplet pool that can oxidize SYR or PTA. Since almost all triplets can transfer energy to dissolved oxygen to make $^1O_2*$, we estimate the quantum yield of total $^3C*$ as $\Phi_{1O2*}/f_\Delta$, where $f_\Delta$ is the fraction of $^3C*$ interaction with dissolved oxygen that forms $^1O_2*$, estimated as 0.53 (Kaur and Anastasio, 2018; McNeill and Canonica, 2016). Therefore, the fraction of triplets that are oxidizing can be calculated as $\Phi_{3C*}/(\Phi_{1O2*}/f_\Delta)$, with values shown in Fig. S26. For $^3C*$ determined by SYR, the fraction ranges from 0.14 to 0.81, with an average of 0.47 ($\pm$ 0.20) and no statistical

difference among the four sample types. This average value is similar to those determined in fog waters (0.55 $\pm$ 0.44) as well as in





previous Davis winter particle extracts (0.31 ± 0.11) (Kaur and Anastasio, 2018; Kaur et al., 2019), indicating that roughly half of the triplets in Davis PM and fog samples are oxidizing. For strongly oxidizing triplets determined by PTA, the fraction ranges from 0.07 to 0.45, with an average of 0.24 (± 0.09); this is half the SYR value, suggesting that approximately half of oxidizing $^3C*$ possesses a high reduction potential, consistent with the results of Fig. 6. For $^3C*$ determined by PTA, Sum-Fall has a statistically

lower average value, 0.11 (± 0.05), compared to Win-Spr (0.29 ± 0.09), FBB (0.22 ± 0.04), and ABB (0.23 ± 0.06). This is reasonable because Sum-Fall samples were not significantly influenced by biomass burning, leading to a lower aromatic content and more weakly oxidizing triplets (McNeill and Canonica, 2016).

### 3.4.4. Quantum yields in aerosol liquid water

We calculated the quantum yields above for the relatively dilute conditions of our particle extracts, but these results are not necessarily applicable to the more concentrated conditions of aerosol liquid water. This is because the formation rate of each oxidant ($P_{Ox}$) is not necessarily proportional to the concentration factor of the sample, while the light absorption should be proportional; based on Eq. 7, if these factors do not vary in the same way as samples get more concentrated, the quantum yield will vary with concentration. As described by Ma et al. (2023a), as we move from dilute extracts to concentrated particle water

$P_{3C*}$ appears to increase linearly with concentration factor, $P_{1O2*}$ does not, and $P_{OH}$ only does sometimes. This suggests that triplet quantum yields in ALW will be similar to those determined in PME, but that yields for singlet oxygen and hydroxyl radical can be lower in ALW compared to in PME. In each case, care needs to be taken when applying the extract quantum yields from above to more concentrated conditions.

### 3.5 Extrapolation of photooxidant concentrations to aerosol liquid water (ALW) conditions

Particle mass/water mass ratios in our PM extracts range from $10^{-5}$ to $10^{-3}$ µg PM/µg $H_2O$ (Table S1), which are typical for dilute hydrometeors like cloud and fog drops (Hess et al., 1998; Nguyen et al., 2016; Parworth et al., 2017). While the results in dilute extracts are interesting and applicable to cloud and fog chemistry, our goal is to understand photooxidant concentrations for each sample type in aerosol liquid water, which is orders of magnitude more concentrated (typically near 1 µg PM/ µg $H_2O$). Due to the very limited water content of particles, we cannot study this condition directly using our current probe techniques. Instead, our

approach has been to quantify photooxidant kinetics (i.e., formation rates and loss rate constants) in a single PM sample as a function of particle dilution and then extrapolate to ALW conditions (Kaur et al., 2019; Ma et al., 2023a). We do this with our current samples by applying parameters obtained from our recent dilution study of a winter (WIN) and a summer (SUM) PM$_{2.5}$ sample (Ma et al., 2023a). Details about the extrapolations and accompanying parameters are provided in Section S4 and Table S10. Moreover, we take the influence of actinic flux on sample types into consideration by using the average midday $j_{2NB}$ value

for each sample type to normalize photooxidant concentrations to that sunlight condition.

We calculate [$^\bullet$OH] in ALW using the average $P_{OH}$ and $k'_{OH}$ values that were determined from the Davis winter and summer particle extracts in our previous study (Ma et al., 2023a). We do not consider the effect of sample type because we do not observe significant differences in the relationship of [$^\bullet$OH] versus DOC among our four sample types (Fig. 5a). As shown in Fig. S27, the

predicted $^\bullet$OH concentration is relatively constant across drop to particle conditions, with a range of $(6 - 9) \times 10^{-15}$ M. The predicted [$^\bullet$OH] in dilute condition is higher than our measured values because we include $^\bullet$OH from the gas phase in our calculation (Kaur et al., 2019). As shown in Figure 8, [$^\bullet$OH] at 1 µg PM/µg $H_2O$ has a range of $(8.8 - 13) \times 10^{-15}$ M, of which the difference among



sample types is driven by the seasonal variation in actinic flux. Our $\bullet$OH concentrations are around 10 times higher than the previous ALW value predicted by Kaur et al. (2019).


We next consider singlet oxygen. As shown in Fig. S29, [$^1O_2^*$] for each sample type increases with particle mass/water mass ratio under dilute conditions, peaks near 0.01 – 0.1 µg PM/µg H$_2$O, and then decreases under more concentrated conditions. At 1 µg PM/µg H$_2$O, Win-Spr has the highest [$^1O_2^*$] (8 × 10$^{-12}$ M), followed by Sum-Fall (3 × 10$^{-12}$ M), FBB (2 × 10$^{-12}$ M), and ABB (1 × 10$^{-12}$ M) (Fig. 8). Win-Spr is characterized by its high $^1O_2^*$ quantum yield, second highest light absorption, and low rate of DOC

quenching for both $^3C^*$ and $^1O_2^*$. In contrast, FBB and ABB have more brown carbon (and therefore greater sources of $^1O_2^*$) but high DOC, which leads to greater sinks for triplets and singlet oxygen. Moreover, DOC in FBB and ABB quenches $^3C^*$ more efficiently than that in Win-Spr (i.e., the BB samples have higher values of $k_{3C^*+DOC}$). Therefore, their [$^1O_2^*$] in ALW are similar to, or even lower than, [$^1O_2^*$] measured in FBB and ABB extracts, while the ALW singlet oxygen concentrations for Win-Spr and Sum-Fall are nearly 20 times higher than their corresponding averages in extracts. Our estimated [$^1O_2^*$] in ALW is 20 – 200 times

lower than the value derived by Kaur et al. (2019), 1.6 × 10$^{-10}$ M, for Davis winter particle water. This is primarily because we account for DOC suppressing $^3C^*$ concentrations, and therefore lowering the rate of $^1O_2^*$ production at high DOC values; this was not done in the previous work.

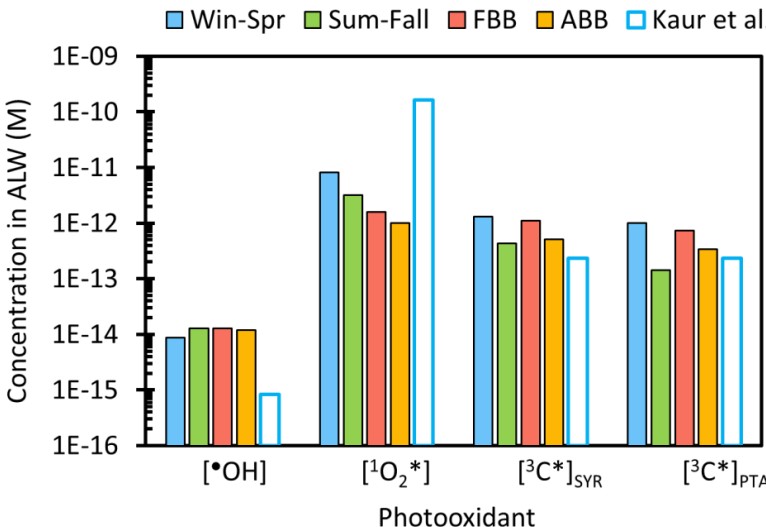

**Figure 8. Predicted photooxidant concentrations for each sample type under aerosol liquid water conditions (1 µg PM/µg H$_2$O), normalized to the average midday actinic flux for each sample type. Previous extrapolations made from Davis winter particle extracts are in open bars, where photooxidant concentrations are normalized to Davis winter solstice sunlight and $^3C^*$ is the lower-bound estimate (Kaur et al., 2019). Photooxidant concentrations all normalized to Davis winter solstice sunlight are in Figure S31.**

Our final ALW predictions are for oxidizing triplets. [$^3C^*$] for all sample types increases with particle mass concentration under

dilute conditions, but then reaches a plateau as solutions become more concentrated and DOC becomes the dominant sink for triplets (Fig. S30). As shown in Fig. 8, [$^3C^*$]$_{SYR}$ and [$^3C^*$]$_{PTA}$ at 1 µg PM/µg H$_2$O have a range of (0.4 – 13) × 10$^{-12}$ M and (0.1 – 10) × 10$^{-12}$ M, respectively, with Win-Spr and Sum-Fall having the maximum and minimum values, respectively. Sum-Fall samples might contain a lower fraction of carbonyl or ketone compounds compared to other sample types, leading to lower production of oxidizing $^3C^*$ (McNeill and Canonica, 2016). Compared to our average measured $^3C^*$ concentration in the PM extracts, [$^3C^*$] in

ALW for Win-Spr and Sum-Fall samples increases by a factor of approximately 20, while ALW concentrations for FBB and ABB



are only around 2 times higher than their extract values. Our predicted $[^3C^*]_{SYR}$ is 2 – 5 times higher than the lower-bound estimate of Kaur et al. (2019) (Fig. 8).

Finally, to understand how photooxidants affect the fate of organic compounds in ALW, we revisit the Kaur et al. (2019) estimates for the lifetimes and fates of five model organic compounds: (1) syringol, (2) methyl jasmonate, (3) tyrosine, (4) 1,2,4-butanetriol, and (5) 3-hydroxy-2,5-bis(hydroxymethyl)furan. To estimate the fate of each compound, we assume equilibrium gas-aqueous partitioning in an aerosol with an ALW of 20 μg m$^{-3}$ and consider reactions with two gas-phase oxidants ($^\bullet$OH, O$_3$) and four aqueous-phase oxidants ($^\bullet$OH, O$_3$, $^3C^*$, $^1O_2^*$). In our calculations, we employed rate constants and Henry's law constants ($K_H$) from Kaur et al. (2019) and used our predicted ALW photooxidant concentrations in Win-Spr ($[^\bullet OH] = 7 \times 10^{-15}$ M, $[^1O_2^*] = 7 \times$

$10^{-12}$ M, $[^3C^*]_{SYR} = 1 \times 10^{-12}$ M, normalized to Davis winter solstice). More details about the calculations are in Kaur et al. (2019). As shown in Figure 9, compounds (1) and (2), which have low $K_H$ values, partition negligibly to the aqueous phase and so gas-phase reactions dominate their fates, with overall lifetimes of 2-3 h; these results are the same for both the aqueous oxidant concentrations of Kaur et al. (2019) and those determined in this work (i.e., Figure 8). For compounds (3), (4), and (5), which have high $K_H$ values, 30 – 100% of the species are present in the aqueous phase of the aerosol. With photooxidant concentrations

predicted by Kaur et al. (2019), organic lifetimes range from 0.04 to 20 h and $^1O_2^*$ is the major sink. However, in this work we predict higher $^\bullet$OH and $^3C^*$ concentrations but significantly lower $^1O_2^*$ in ALW (Figure 8). The lower $^1O_2^*$ leads to lifetimes of compounds (3) and (5) increasing by factors of 6 and 17, respectively. $^3C^*$ becomes the dominant oxidant for the phenolic amino acid, compound (3), but singlet oxygen is still the dominant sink for the substituted furan, compound (5). With the new oxidant concentrations, the lifetime of the aliphatic alcohol, compound (4), decreases by a factor of almost 3 due to the enhanced $^\bullet$OH

concentration and singlet oxygen is much less important. Overall, results with the new oxidant concentrations show some significant shifts in the lifetimes of the three highly soluble organics as well as in the contributions of individual oxidants. But our new results still indicate that $^3C^*$ and $^1O_2^*$ dominate the particle processing for highly soluble organic compounds with which they react quickly while $^\bullet$OH dominates for aqueous organics that react slowly with the other two oxidants. Based on our Win-Spr oxidant concentrations (Figure 8), for an organic compound that has an $^\bullet$OH rate constant of $1 \times 10^{10}$ M$^{-1}$ s$^{-1}$, singlet oxygen will

be the dominant oxidant if its rate constant with the organic is larger than roughly $1 \times 10^7$ M$^{-1}$ s$^{-1}$, while oxidizing triplets will dominate if their rate constant is larger than approximately $1 \times 10^8$ M$^{-1}$ s$^{-1}$,





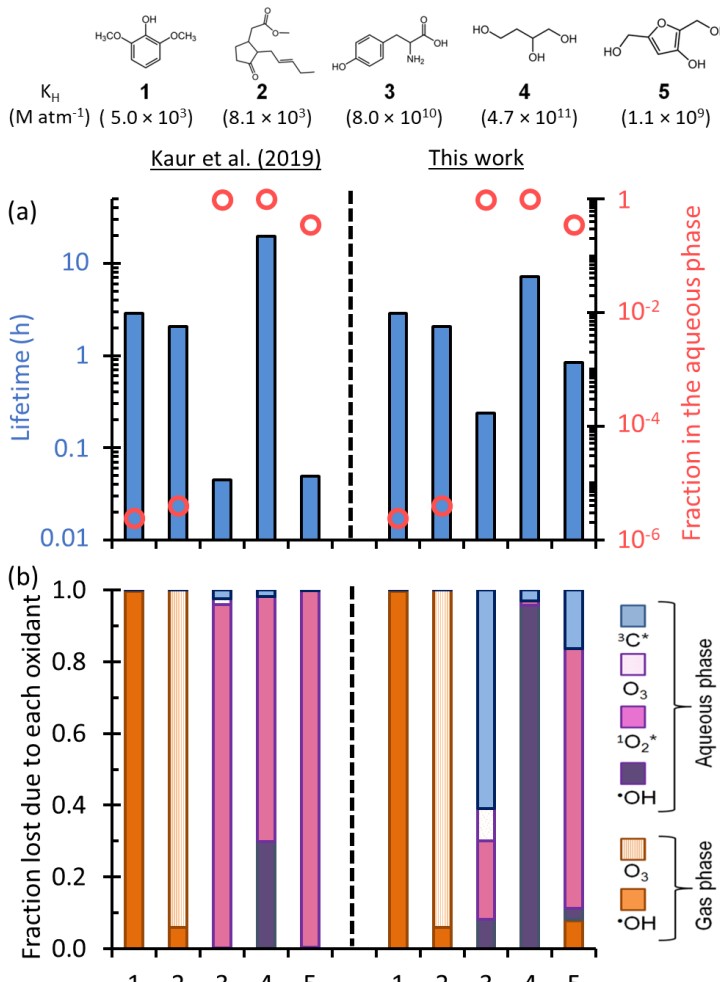

**Figure 9. Fates of five model organic compounds – (1) syringol, (2) methyl jasmonate, (3) tyrosine, (4) 1,2,4-butanetriol, and (5) 3-hydroxy-2,5-bis(hydroxymethyl)furan in an aerosol (20 μg dry PM/m³-air) containing equal amounts of PM and liquid water (i.e., 1 μg PM/μg H₂O). Results to the left of the dashed lines are calculated using estimated photooxidant concentrations from Kaur et al. (2019), while those to the right are calculated using oxidant concentrations for Win-Spr from this work. Panel (a) shows the overall lifetime (blue columns, left *y*-axis) and the fraction in the aqueous phase (red circles) for each organic. Panel (b) shows the fraction of organic lost due to each oxidant.**

## 4. Conclusions, Implications, and Uncertainties

In this work, we measured concentrations of three photooxidants – hydroxyl radical, singlet molecular oxygen, and oxidizing triplet excited states of brown carbon – in particle extracts. Our extracts have particle mass/liquid water mass ratios in the range of $(0.7-9.1) \times 10^{-4}$ μg PM/μg H₂O, which are close to fog/cloud water conditions but much more dilute than aerosol liquid water. We categorized samples into four types based on sampling dates and chemical characterization: Winter & Spring (Win-Spr), Summer & Fall (Sum-Fall) without wildfire influence, fresh biomass burning (FBB), and aged biomass burning (ABB). FBB contains the highest amounts of BrC, leading to the highest average mass absorption coefficients normalized by dissolved organic carbon, e.g., 3.3 (±0.4) m² (g C)⁻¹ at 300 nm. Win-Spr and ABB have similar MACs at this wavelength (1.9 (±0.4) m² (g C)⁻¹ and 1.5 (±0.3) m² (g C)⁻¹, respectively), while Sum-Fall has the lowest MAC$_{DOC}$ (0.65 (±0.19) m² (g C)⁻¹).





Photooxidant concentrations in the particle extracts are in the range $(0.2\text{-}4.7) \times 10^{-15}$ M for $^{\bullet}$OH, $(0.07\text{-}4.5) \times 10^{-12}$ M for $^1O_2^*$, and $(0.03 - 7.9) \times 10^{-13}$ M for $^3C^*$, respectively. All oxidant concentrations generally increase with the concentration of dissolved organic carbon (DOC), which ranged from 5 to 192 mg C L$^{-1}$. $^1O_2^*$ concentrations exhibit good linearity with DOC with all sample types falling roughly on the same line. Fresh BB extracts have the highest $[^1O_2^*]$ but the lowest average quantum yield ($\Phi_{1O2^*}$), while Sum-Fall samples are the opposite. $\Phi_{1O2^*}$ is negatively correlated with MAC$_{DOC}$, indicating that less light-absorbing samples

form $^1O_2^*$ more efficiently. Triplet concentrations determined by both probes linearly increase with DOC, and this relationship for $[^3C^*]_{SYR}$ is independent of sample type. We find that approximately half of the total triplets are oxidizing based on SYR loss, while roughly half of the oxidizing triplets are strongly oxidizing based on PTA loss. FBB has the lowest average $\Phi_{3C^*}$, while atmospheric aging appears to enhance $\Phi_{3C^*}$, as well as $\Phi_{1O2^*}$, based on the higher quantum yields for ABB samples.

Based on our results in dilute PM extracts (as well as past work), light absorption by brown carbon produces significant amounts of photooxidants in particles. To estimate the corresponding photooxidant concentrations, we extrapolate measured photooxidant kinetics in our particle extracts to an aerosol liquid water condition (1 µg PM/µg H$_2$O). Estimated concentrations of $^1O_2^*$, $^3C^*$, and $^{\bullet}$OH in ALW are on the order of $10^{-12} - 10^{-11}$, $10^{-13} - 10^{-12}$ and $10^{-14}$ M with the ratio of $^1O_2^*$: $^3C^*$: $^{\bullet}$OH of $(900 - 90) : (150 - 10) :$ 1. The corresponding ratio in our particle extracts is $(40 - 5) : (1 - 10) : 1$. For Win-Spr and Sum-Fall samples, singlet oxygen and

oxidizing triplet concentrations increase significantly in ALW compared to in dilute extracts, while the changes in FBB and ABB are minor, likely due to the high DOC in the extracts, which cause strong quenching of $^1O_2^*$ and $^3C^*$. Compared to the predicted photooxidant concentrations in Davis winter particle water by Kaur et al. (2019), our Win-Spr predictions for $[^{\bullet}$OH$]$ and $[^3C^*]$ are nearly 10 and 5 times higher, respectively, but our ALW value for $[^1O_2^*]$ is 20 times lower. Based on our estimated ALW concentrations, lifetimes of organic compounds with high Henry's law constants in ALW can be significantly shortened compared

to foggy conditions (Kaur et al., 2019), due to enhanced $^3C^*$ and $^1O_2^*$ concentrations in particle water.

While oxidant concentrations are required to calculate the lifetimes of individual organic species in ALW, the formation rate of a photooxidant provides insight into the overall significance of that oxidant as a sink for organics. Since organic compounds appear to be the major sink for all three photooxidants in ALW, the formation rate of an oxidant is approximately equal to the rate of

DOM processing by that oxidant, although organics can also physically quench a triplet without transforming the organic (Grebel et al., 2011; Ma et al., 2021; Smith et al., 2014). Based on our extrapolations, the ratio of formation rates in ALW for $^1O_2^*$, $^3C^*$, and $^{\bullet}$OH (including mass transfer from the gas phase) is 1: 100: 4, taking Win-Spr as an example. Since the triplet formation rate is much higher than those of $^{\bullet}$OH or $^1O_2^*$, our results indicate that $^3C^*$ might be more important for the overall oxidation of organic compounds compared to the other two oxidants. However, the picture for any specific organic compound depends on its rate

constants with each oxidant. For example, $^{\bullet}$OH will be relatively more important for organics that are less reactive with $^3C^*$ and $^1O_2^*$.

There are important uncertainties in the oxidant concentrations reported in our work. Foremost, predicting photooxidant concentrations from our dilute extracts to ALW conditions is highly uncertain as it requires extrapolating over a concentration

difference of approximately a factor of 1000. While our current extracts have more DOC than those in our past work (Kaur et al., 2019), allowing us to get closer to ALW chemistry, we are still orders of magnitude more dilute. Despite this improvement, additional approaches – such as chamber and flow tube studies – are needed to measure photooxidants and their chemical impacts under conditions more similar to ambient aerosols. The oxidizing triplet concentrations are less certain than those of the other two

oxidants, both because we use an individual triplet ($^3$DMB*) to model the wide range of natural triplet reactivities, but also because of uncertainties in correcting the inhibition of syringol oxidation by particle components. Another uncertainty with our current (and past) results is that we are missing the water-insoluble chromophores from particles. Consistent with past results from other groups, we find significant amounts of highly light-absorbing water-insoluble brown carbon in our particle samples, suggesting that by using aqueous extracts we are underestimating the concentrations and significance of photooxidants in ambient particles. This issue should be addressed in future photochemistry studies.


## Data availability

All data are available upon request.

## Author contribution

CA and LM developed the research goals and designed the experiments. KB lent and set up the sampler, while
LM and CG collected samples. LM, RW, and LH performed the photochemistry experiments while WJ and CN analyzed OC and ions, respectively. LM analyzed the data and prepared the manuscript with contributions from all co-authors. CA reviewed, wrote portions of, and edited the manuscript. CA and QZ provided supervision and oversight during the experiments and writing.

## Competing interests

The authors declare that they have no conflict of interest.

## Acknowledgement

We gratefully acknowledge the following agencies for their publicly available data: the California Air Resources Board for PM$_{2.5}$ data, NASA's Land, Atmosphere Near real-time Capability for EOS (LANCE) system (https://earthdata.nasa.gov/lance), part of the Earth Observing System Data and Information System (EOSDIS) for wildfire and smoke images, and the NOAA Air Resources Laboratory (ARL) for the HYSPLIT model and READY website (https://www.ready.noaa.gov).

## Financial support

This research has been supported by the National Science Foundation (AGS-1649212 and AGS-2220307); the California Agricultural Experiment Station (Projects CA-D-LAW-6403-RR and CA-D-ETX-2102-H); and the University of California, Davis (Donald G. Crosby Graduate Fellowships in Environmental Chemistry and Jastro Shields Research Awards).




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
