# Peer review of "Seasonal variations in photooxidant formation and light absorption in aqueous extracts of ambient particles"

_EGUsphere, 2023_

## Author Response (AR1)

**Response to Reviewers**

"Seasonal variations in photooxidant formation and light absorption in aqueous extracts of ambient particles" by Lan Ma et. al.

Each reviewer comment is listed in italics and our response, in plain text, is directly below it. Line numbers in the revised version are different from the original (e.g., in the reviewers' comments) due to changes in the manuscript.

*Anonymous Referee #1

*The manuscript aims at measuring the seasonal variation in photooxidant formation and concentration in atmospheric water and to predict the lifetime of 5 compounds in the atmosphere. Overall, I found the article well written and would support its publication as it brings interesting information to the community.*

*I did not find major issues in the article, here is my list of comments and corrections:*

*Abstract and introduction*

*The abstract and introduction are clear. In addition to singlet oxygen, excited triplet states and hydroxyl radical, the authors could also mention in the introduction other photooxidants that were not considered in the study but that may play a role in the transformation of some classes of contaminants. E.g., Halides radicals may play a role in the transformation of electron rich compounds (Marine Chemistry 115 (2009) 134–144) or long-lived photooxidant could be important for the transformation of phenols or anilines (Water Research 213 (2022) 118095).*

**Response**: We thank the reviewer for their thoughtful review and encouraging comments. While adding descriptions of other oxidants has some merit, we haven't done this since there are so many other photooxidants (e.g., hydrogen peroxide, reactive halogens, hydrated electrons, superoxide, hydroperoxyl radicals, and sulfate radicals) and our focus is on the three oxidants we highlight in the introduction. In addition, the paper is already quite long so we're reluctant to add more text.

*L25. It looks to me that the OH quantum yield value is too high and does not correspond to the values presented in the article (Table S3).*

**Response:** Thank you for pointing this out. We had two connected mistakes in our quantum yields for hydroxyl radical. First, the multiplier on the $^\bullet$OH quantum yields in the column heading of Table S3 was written as $10^{-4} \times \Phi_{OH}$, but it should have been $10^4 \times \Phi_{OH}$. Second, in the abstract we listed the average $^\bullet$OH quantum yield as 3.7 ($\pm$ 2.4) %, but it should have been 0.037 ($\pm$ 0.024) %. We have corrected both errors.

*L.79. I would switch organic compounds for DOM as the quoted studies presents correlations between 3DOM\* quantum yields and factors correlating with the molecular weight / aromaticity.*

**Response:** Thank you for your suggestion. We reworded the sentence.

*Material and methods*

*L.141. I would indicate the spectrophotometer cuvette pathlength.*

**Response:** Thank you for your suggestion. We added this information.

*I.146 I would add in the SI the arc lamp spectra, that is important to evaluate nitrate photolysis.*

**Response**: The lamp spectrum is in our previous work, so we decided not to include it in this work. We added the information that the spectra of simulated sunlight can be found in our previous paper.

*Results and discussion*

*The results are presented in a logical order, I have two main comments on the results:*

1. *Hydroxyl radical quantum yields are presented. The fact that hydroxyl radicals are produced by many pathways in the atmospheric aqueous phase, and that each pathway has its own quantum yield, makes the numbers difficult to compare to other studies and not that useful. The quantum yield numbers would depend on the extract's composition (nitrate, nitrite, iron) but also on the irradiation wavelength distribution.*

   **Response:** The quantum yields we determined are apparent quantum yields, based on the overall rates of OH formation and light absorption by the particle extracts. •OH production from any given chromophore (e.g., nitrate photolysis) has its own quantum yield and our apparent quantum yield values are generally lower.  That information is, by itself, somewhat useful.  But the apparent quantum yield is even more useful for calculating an oxidant production rate for a natural sample under a known photon flux. We added a statement about apparent quantum yields to Section 3.4.1.

2. *Part 3.5. It looks like the authors use Henry constants to evaluate the partition of 5 compounds between the atmospheric aqueous phase and the gas phase. The use of Henry constants is fine for dilute solutions, but I fear that for concentrated solution (1ug PM/ug H2O), the actual partition may be different from the one calculated using Henry constants. I think that the authors should at least acknowledge the problem. If the authors are aware of methods or measurements to evaluate the actual partition coefficients to use them instead of Henry constants.*

   **Response:** We agree that Henry's law might not work well under concentrated conditions, which would affect the partitioning. For example, phenols can be "salted out" by high concentrations of salts, so that under ALW conditions the extent of partitioning to the salty aqueous phase can be lower than compared to nearly pure water (McFall et al., 2020).  But this behavior (and the corresponding activity coefficient of the organic compound) depends strongly on the salt identity, with some salts having a large impact

and others a minor effect.  Given this complexity, we used the dilute Henry's law constant to get a rough sense of organic reactivity, and the importance of different oxidants, in the gas and aqueous phases.  We have modified the text to acknowledge this issue.

*Figures, the date format may confuse non-American reader (e.g., one can read the first date as November first 2019 or January 11ᵗʰ 2019). I would suggest writing the months to be clearer. Also, the numbers on the y-axis could be written as $1{\times}10^{-15}$ (and not 1E-15).*

**Response:** Thank you for your suggestions. We now describe the date nomenclature in the captions to remove the ambiguity.  Our plotting software makes it difficult to use formal exponential notation, so we have left our exponents as they were.

*L.306. "fresh BB are fragmented during aging", it could be noted that ozone exposure also induces and increase of E2/E3 (Leresche et al. quoted in the manuscript) and that ozone indeed also induce a decrease in mean molecular weight indicating that fragmentation occurs during ozonation (Environmental Science & Technology, 2023 57 (14), 5603-5610).*

**Response:** We agree that ozone fragments organic compounds and have added these references to the main text.

*L.347. DDT assay, the abbreviation is not defined, switch for the full name.*

**Response:** Thank you for your suggestion. We added the full name of DTT in the main text.

*L.450. Do the authors think that there are anilines moieties in PME? I would suggest withdrawing the mention to anilines.*

**Response:** We agree aniline moieties are likely minor constituents in atmospheric particles, especially compared to phenols. We have deleted aniline from the description.

*L.508. The second-order rate constant between singlet oxygen and water was reevaluated to be of $2.76{*}10^5$ $M^{-1}$ $s^{-1}$ (Environ. Sci.: Processes Impacts, 2017, 19, 507–516) I would suggest using the more recent value.*

**Response**: Thank you for your suggestion. We decided to stick with the rate constant we originally used ($2.2 \times 10^5$ $s^{-1}$) because we also used this value for our previous work and would like to be consistent between these papers. If we had changed to the newer, Appiani et al. (2017) rate constant, $^1O_2^*$ production rates and quantum yields would increase by 26%, while $^1O_2^*$ concentrations would decrease by 3%. We have added mention of this new rate constant, and the impact of the rate constant difference, to Section 3.4.2.

*L.552. $^3C^*$ fraction that produces singlet oxygen (fΔ). This fraction was recently measured for Suwannee River fulvic acid to be of 0.34 (Environ. Sci. Technol. 2017, 51, 13151−13160). The value from McNeill and Canonica is a rule of thumb I believe. It would be worth mentioning this 0.34 value.*

**Response:** Thank you for pointing out this $f_\Delta$ value, which we have added to the text. However, since Schmitt et al. (2017) determined this value based on the $^3C^*$ quantum yield at 346 nm, and since this quantum yield can vary with wavelength and DOM type, we have stuck with our original estimate of $f_\Delta$ for our calculations.

*L.678. "Estimated concentrations of $^1O_2$, $^3C^*$, and OH in ALW are on the order of 10-12 - 10-11, 10-13 - 10-12 and 10-14 M". I would suggest putting the respective number range next to the corresponding reactive species, as it is, it is difficult to see which numbers correspond to what.*

**Response:** Thank you for your suggestion. We made this change.

*L.993 and L.66, it should be Hoigné and not Hoigne.*

**Response:** Thank you for your suggestion. We have corrected this.

***Anonymous Referee #2

*Overview:*

*The authors of this manuscript present OH, 3C* and 1O2* measurements of 18 filters taken from Nov 2019 to Oct 2020 in Davis already described and published in (Jiang et al., 2023). In Jiang et al., the concentrations of OH, 3C* and 1O2* are presented for each filter in Figures 5, 6, S11, S12.*

*The authors of this manuscript present MAC values for their extracts, the same values as in (Jiang et al., 2023). They also discuss the AMS data from (Jiang et al., 2023). The quantum yields are also discussed in (Jiang et al., 2023). Finally, the authors extrapolate the OH, 3C* and 1O2* concentrations to aerosol liquid water content, which they already did for 2 of the same samples in (Ma et al., 2023b).*

*Therefore, this paper is not publishable as all the data has been previously published across two papers by the same authors: (Ma et al., 2023b; Jiang et al., 2023).*

**Response:** We thank the reviewer for their comments, but we strongly disagree with their characterization of these three papers. We also strongly disagree with their incorrect assessment that "all" (or even a significant fraction of) the results from the current manuscript have already been published. However, we see now that we hadn't sufficiently explained the connections and differences between these three manuscripts in the current work. To remedy this, we added several sentences to the end of the Introduction. Below we have included a more detailed version of this addition.

The main purpose of Ma et al. (2023) (which is equivalent to Ma et al. (2023b) in the reviewer's references) was to understand how to extrapolate measurements of photooxidant concentrations in relatively dilute particle extracts to the much more concentrated conditions of aerosol liquid water (ALW). We use this approach because there is currently no way to directly measure oxidant concentrations in ALW. Following our method in Kaur et al. (2019), where we studied a single winter PM sample under different dilutions, in Ma et al. (2023) we studied both

a winter and summer (wildfire) sample under multiple dilutions.  We measured oxidant concentrations in each dilution, which allowed us to derive the oxidant kinetics (production rates and loss rate constants) that we needed to predict oxidant kinetics and concentrations in ALW for our two PM samples.

In our current manuscript, we measure concentrations of $^{\bullet}OH$, $^{3}C^*$, and $^{1}O_2^*$ in 16 PM samples that we collected over the course of a year, each studied at one extract dilution.  We then apply the oxidant kinetics determined in Ma et al. (2023) to our new extract measurements to estimate oxidant concentrations in aerosol liquid water for the 16 samples. As part of this, we also include the concentration results for the 2 samples from Ma et al. (2023) to make the dataset more complete.  The focus of our current manuscript is to study how photooxidant formation varies with season and biomass burning (BB) influence.  This is important because there are no previous studies on $^{3}C^*$ seasonality and only a few studies on the seasonality of $^{1}O_2^*$ or $^{\bullet}OH$ in PM extracts.

The third paper in this trio, Jiang et al. (2023), was headed by Dr. Wenqing Jiang in Professor Qi Zhang's group, with Drs. Ma and Anastasio as co-authors. The main goal was to use positive matrix factorization (PMF) to resolve the year of PM extracts (i.e., the samples that we studied in the current manuscript) into different organic aerosol (OA) types and examine the ability of each OA factor to form oxidants. Jiang et al. (2023) used PMF on two characteristics of the extracts: composition as determined by aerosol mass spectroscopy (AMS) and UV/Vis measurements. The PMF results indicated that there were five different OA factors: fresh BB, aged BB, and three oxidized organic aerosol (OOA) types. They then combined the PMF results with our oxidant measurements to predict oxidant production potentials ($PP_{OX}$) for each of the five OA factors, which they then used to estimate oxidant concentrations under cloud/fog conditions for past AMS field studies. While the Jiang et al. paper is related to our current work (and shares the same dilute extract oxidant data), our current manuscript is markedly different: we focus on the measured oxidant results and delve deeply into how oxidant kinetics are related to non-AMS sample characteristics (e.g., E2/E3 and DOC concentrations).  In contrast to the Jiang et al. work, in our current manuscript we say almost nothing about the AMS characteristics, the five OA factors, or the factor $PP_{OX}$ values, which are the heart of Jiang et al. (2023).

We hope that this explains how the three papers are connected but also separate and complementary.

As for the specific criticisms of overlap above, here are our responses:

*Reviewer: The authors of this manuscript present OH, 3C* and 1O2* measurements of 18 filters taken from Nov 2019 to Oct 2020 in Davis already described and published in (Jiang et al., 2023). In Jiang et al., the concentrations of OH, 3C* and 1O2* are presented for each filter in Figures 5, 6, S11, S12.*

**Response:** Oxidant measurements for our 18 extracts (16 from the current manuscript and 2 from Ma et al. (2023)) are presented in Figure 5a of Jiang et al. (2023).  But the main points of the Jiang et al. figure are ideas we do not discuss in our current manuscript: (1) the relative significance of the five PMF factors for oxidant generation and (2) the good agreement between the sum of the PMF-derived oxidant concentrations and the measured oxidant concentrations in each sample.  The text of Jiang et al. only mentions the average concentration for each of the

three oxidants, while our current manuscript gives much more detail about the extract oxidant concentrations.

Figure 6 of Jiang et al. does not give oxidant concentrations in the PM extracts. Rather, it presents derived $PP_{OX}$ values, i.e., the ability of each of the five PMF-derived OA factors to make each oxidant. This is not a topic we discuss in our current manuscript.

Figure S11 of Jiang et al. shows the contributions of the five OA factors to the extract concentrations of $^{\bullet}OH$, $^3C^*$, and $^1O_2^*$. This is not a topic we discuss in the current manuscript. This figure also reiterates the average extract concentration for each of the three oxidants.

Figure S12 of Jiang et al. does not show oxidant concentrations. It shows correlation coefficients between oxidant concentrations and different AMS ion families or AMS tracer ions. The goal was to try to identify elemental or molecular components that are associated with oxidant generation. We do not discuss this idea in our current manuscript.

*Reviewer: The authors of this manuscript present MAC values for their extracts, the same values as in (Jiang et al., 2023). They also discuss the AMS data from (Jiang et al., 2023). The quantum yields are also discussed in (Jiang et al., 2023). Finally, the authors extrapolate the OH, 3C\* and 1O2\* concentrations to aerosol liquid water content, which they already did for 2 of the same samples in (Ma et al., 2023b).*

**Response:** The MAC values in our current manuscript are the average values for each of the four sample types that we qualitatively determined (fresh BB, aged BB, Winter-Spring, and Summer-Fall). The MAC values in Jiang et al. are the PMF-derived values for the five identified OA types. There is good agreement across these two classification schemes, which is a positive development that we now point out. We appreciate that the differences in the two sets of MAC are confusing, so we also added some text to the discussion of Figure 2 to explain this.

As for overlap with the AMS data, the Jiang et al. paper goes into great detail about the AMS analysis of the PM extracts, including major PM components as determined by AMS (OA and inorganic ions), atomic ratios for the OA (O/C, H/C, N/C), AMS mass spectra, and AMS organic tracer ions. In contrast, we say almost nothing about the AMS results in our current manuscript: in Section 3.1 we give two sentences indicating that the AMS-PMF results confirm our qualitative assignments for the fresh and aged BB particle samples.

In contrast, we extensively discuss oxidant quantum yields in our current manuscript but included very little about them in Jiang et al. (2023). As best we can tell, there is only one sentence about quantum yields in Jiang et al., on page 1115: "This suggests that the BrC chromophores in $_{WS}BBOA_{fresh}$ are less efficient sources of $^1O_2^*$ and $^3C^*$, i.e., have lower quantum yields.[48]" This reference 48 in Jiang et al. is our current ACP manuscript, so there is no real overlap here.

The final point in the reviewer's comment above is that our current paper extrapolates aqueous extract oxidant concentrations to ALW conditions, but that we previously did this for the 2 samples in Ma et al. (2023). Yes, this is true. But we see it as leveraging our past work rather than a problem: we applied the kinetics that were painstakingly determined from the 2 samples in Ma et al. (2003) to the 16 extracts that we studied in our current manuscript to predict their ALW oxidant concentrations.

*Reviewer: Therefore, this paper is not publishable as all the data has been previously published across two papers by the same authors: (Ma et al., 2023b; Jiang et al., 2023).*

**Response:** As we detail above, very little of the data in our current manuscript was previously published in Ma et al. (2023) or Jiang et al. (2023). Ma et al. (2023) measured oxidant formation in two samples, each studied at multiple dilutions, in order to understand how to extrapolate to ALW conditions. Our current work focuses on seasonal variations in oxidant concentrations in dilute solution, delves deep into the details of the oxidant results, and uses the kinetic results from Ma et al. (2003) to predict ALW oxidant levels. Jiang et al. (2023) focused on AMS analysis of the extracts, using PMF to discern different OA factors, and then used our oxidant measurements to estimate oxidant reactivities for the five PMF-determined OA factors. While these three papers are connected and complementary, each has a separate focus and a different set of main points. We have added several sentences at the end of the introduction to better explain the links and differences between these three distinct pieces of work.

*Comments:*

*Nevertheless, the techniques used, although uncommon in the community (like use of D2O for FFA, use of double probe for 3C\* - although that's building on their own previous work in (Ma et al., 2023a) which has interesting merit -, acidifying to pH4.2 with no clear understanding of the impact of pH), have been reported in other publications by the same authors. The data are listed in tables in the SI in a good and extensive matter (but missing LOD info). Unfortunately, there is no new key message or finding in this submitted manuscript in comparison to previously published work by the same group, and the paper has important issues that would need to be resolved.*

**Response:** This comment seems to criticize our use of techniques that were developed previously, but to us this is a natural progression: develop methods and then apply them to ambient samples. We did not examine the impact of pH on oxidant generation, so this would be a good topic for a future paper. As for the comment that we have no new key messages, we disagree: we identified multiple key points about oxidant levels and generation in the manuscript, but will highlight just two here. First, Figure 8 shows our sample-type average oxidant concentrations in ALW and a comparison to the winter results in Kaur et al. (2019), which was the first time we used dilute extract measurements to estimate oxidants in particle water. Our new results show significant differences with the Kaur work: we find [OH] is approximately 10 times higher, $[^1O_2^*]$ is roughly 100 times lower, and $[^3C^*]$ is 2-5 times higher. These important differences result in significant changes in the lifetimes and fates of organic molecules, which brings us to a second key message: as we show in Figure 9, $^1O_2^*$ is much less important that we predicted in Kaur et al. (2019), while triplets and OH are more important than previously estimated.

*General issues with this paper beyond the lack of new data/results are listed here:*

- *Raw data of all the BA, FFA, SYR and PTA probe decays for all the samples is missing.*
  - *There is one example of the BA decay which for the 121719 and the 030420 samples is clearly not linear. This observation is concerning as the deviation from linearity indicates that the oxidant is no longer under pseudo-first order rate kinetics! What do the probe kinetics look like for other oxidants and other filters?*

**Response:** FFA, SYR, and PTA decay in approximately 90% of samples were pseudo first order. For BA, over 60% of the samples followed pseudo first order decay. We specifically included Figure S1 (now S2) to highlight samples where probe decay was not first order. We do not believe this is concerning but rather a case where a small pool of more reactive OH-producing species gives a higher initial burst of OH, followed by a sustained and lower OH steady-state concentration as the less-reactive species drive OH production. This phenomenon has been described previously, as we cited in the text.

We have added a figure to the supporting information (new Figure S1) to show examples of the raw data of BA, FFA, SYR, and PTA decay in a PM extract. As shown in this figure, for most of our experiments probe decay was pseudo first order, indicating oxidant concentrations were at steady state. Therefore, using probes to determine oxidant concentrations in our sample is a reliable method. We added this information, and an overview of the percent of samples that showed first-order probe decay, in Section 2.3.

- *A number of incorrect statements are used to motivate the study, often based on "things being unknown". Here are examples:*

  - *Lines 68-69: So much is known about measured and modeled OH radical concentrations in the gas phase and its seasonality (Martin et al., 2003; Fan and Li, 2022) and so simply by partitioning, one could estimate what the seasonality might be (I would agree with a statement about OH radical concentrations being variable due to different sinks, but the word "unknown" is a disservice to the OH radical community (ex: Comprehensive OH seasonality by (Pfannerstill et al., 2021) and OH has been quantified at the global scale: (Thames et al., 2020) and (Pimlott et al., 2022) are examples.*

    **Response:** While mass transport of gas-phase •OH can be a dominant source of hydroxyl radical to cloud and fog drops, it is a minor source to particle water (Ma et al., 2023). Thus the seasonality of •OH in aqueous aerosol cannot be predicted based on the known seasonality of gas-phase •OH. We have clarified this in the introduction.

- *References are an issue throughout the text where multiple papers (5-6) are referenced without identify the contribution of each and thereby missing the opportunity to build upon previous work. Here are a few examples to support this claim:*

  - *Lines 53-55: 6 seemingly random references are listed to support the fact that OH, 3C* and 1O2* are important oxidants. Reviews such as (McNeill and Canonica, 2016; Ossola et al., 2021; Hems et al., 2021) are more appropriate*

  - *Statement on Lines 98-99 is inaccurate as (Bogler et al., 2022) addresses both the seasonality and the particle type.*

  - *Line 263: a study from 2001 and from 2013 were chosen to discuss organic carbon content in biomass burning, when there are more recent references: to name a few: (Fang et al., 2023; Di Lorenzo et al., 2017; Lee et al., 2016; Bikkina and Sarin, 2019; Forrister et al., 2015)*

o *Same point is true for line 281-283 where the 4 references listed are not representative of the statement, see for example (Fleming et al., 2020; Lee et al., 2014; Laskin et al., 2014)*

o *Another example on lines 284-285*

**Response:** Thank you for the comment. We have revised the text based on the suggestions, with two exceptions: (1) on Line 263, we are specifically citing the OC/PM2.5 ratio of BB particles, not just the organic matter content or O/C ratio and (2) some of the Line 281-283 references that reviewer recommended are not applicable to the points we are making in the text.

- *The authors chose to focus on a seasonality story line, but was 2020 representative? There were massive wildfires in Fall 2020 in northern California.*

    o *Where did the PM2.5 data in Figure 1 come from? (I found it at the bottom of Table S1 in footnote b…but it should be in the text and appropriately referenced with multiyear data)*

    **Response:** Thank you for pointing this out, we have added the data source to the caption of Figure 1.

    o *What is the seasonal PM2.5 profile in northern California? Was 2020 representative of PM mass?*

    **Response:** According to data from the California Air Resources Board, Davis typically has the highest $PM_{2.5}$ concentrations in the winter due to residential wood combustion and lower $PM_{2.5}$ in summer. But there can also be summer peaks, whether from multi-day stagnant subsidence inversion episodes or wildfires. Certainly 2020 had massive summer wildfires, but this reflects the increasing summer wildfire activity in the west. The bottom line is that "representative" is changing.

    o *The methods sampled PM10 – how different/similar are PM10 to PM2.5 in Davis.*

    **Response:** We sampled $PM_{2.5}$, not $PM_{10}$. While the sampler has a $PM_{10}$ head, this is followed by two slotted plates to remove particles greater than $PM_{2.5}$ prior to collection on a filter. We modified the text to clarify this.

- *There were no samples taken between March 4th 2020 and July 7th 2020 (Table S1) and there are therefore no spring samples. The use of spring seasonality is therefore unjustified throughout the text.*

    **Response:** In Davis, February is typically the beginning of spring from a temperature perspective. For example, in February 2020, the average high temperature was around 16 °C, while the highest high was 24 °C. The high on March 4th 2020 was also 24 °C. Therefore, we regard some of the February and March samples as more representative of spring conditions than winter.

- *The authors motivate their work discussing Fenton OH chemistry (lines 61-64) but how do they take this chemistry into account in their own measurements of OH steady state concentration calculations?*

   **Response:** We don't use Fenton chemistry to motivate our work but do include it in this section as one of the likely sources of $^{\bullet}OH$. Our measurements determine the overall steady-state concentration of hydroxyl radical but do not give us much insight into the sources.

- *Relevant work that should have been built upon to connect to ROS and EPFRs (also from ambient Californian samples): (Fang et al., 2023)*

   **Response:** While there are likely interesting connections between photochemically formed oxidants and the biologically focused reactive oxygen species (ROS) and environmentally persistent free radicals (EPFR) in PM samples, this is beyond the scope of our current work. It is difficult to connect our photooxidant measurements with those of ROS and EPFR. In part this is because ROS and EPFR are measured in the dark, sometimes under physiologically relevant conditions, while we measured photochemically-produced oxidants under atmospherically relevant conditions.

- *No mention of limits of detection. What are the minimum concentrations that the authors are able to quantify (3 sigma above background)?*

   **Response:** We didn't provide limits of detection because, unlike standard analytical measurements, there are not clear LODs for probe methods. Generally, oxidant measurements using probes quantify the decay of probe as a function of illumination time. For dilute samples with low oxidant concentrations, we simply extend the illumination time to obtain a quantifiable probe decay.

- *The authors decided to divide their concentrations by 7 for comparing filters collected for 7 days and filters collected for 1 day. This division is an oversimplification of the complex mixture of brown carbon and is not justified.*

   **Response:** For $^{1}O_2^*$ and $^{3}C^*$, dividing the 7-day filter sample concentrations by 7 works well because under our dilute extract conditions these oxidant concentrations are linearly related to the particle mass/water mass ratio; we showed this in our previous work (Kaur et al., 2019; Ma et al., 2023). As for $^{\bullet}OH$, we did not simply divide its concentration by 7 since the relation between $^{\bullet}OH$ concentration and particle mass/water mass ratio is not as straightforward. Therefore, as described in the text, we fitted $^{\bullet}OH$ concentration to particle mass/water mass ratio with a linear regression and used this fitting to estimate the normalized $^{\bullet}OH$ concentrations with the time-normalized particle mass/water mass ratio values.

- *Line 15: The abstract mentions that; "there are few measurements of these photoxidants…" which is not accurate. There are likely over a dozen: (Faust and Allen, 1992; Anastasio and McGregor, 2001; Albinet et al., 2010; Hong et al., 2018; Cote et al., 2018; Manfrin et al., 2019; Kaur et al., 2019; Leresche et al., 2021; Jiang et al., 2023; Bogler et al., 2022; Lyu et al., 2023; Ma et al., 2023b)!*

**Response:** Many of these studies only measured •OH, while relatively few measured $^1O_2^*$ or $^3C^*$. We have revised the text to clarify this.

- *The mathematical equations representing the projected concentrations in AWL are missing.*

    **Response:** Thank you for your suggestion. We have added an equation in Section 3.5 to show the basic relation for oxidant concentration prediction in ALW.  The two pieces of this simple equation – i.e., the relationships between oxidant production rate and DOC (as a marker for concentration factor) and oxidant loss rate constant and DOC – are in Table S10.

- *Presentation of wildfire information in lines 231-236 but making no connection to the oxidant data.*

    o   Wouldn't a discussion on the different BBOA samples have been more worthwhile for the community?

    **Response:** Thank you for your suggestion. The wildfire information here is to provide a basic background information and we connected the information to the sample absorption, composition, and oxidant data later. We have added information on how the plume transport time affects particle sample type.

- *There is considerable research undertaken to study the impact of solvent extraction on filters that the authors should be building upon: (Chen et al., 2022) and references therein. (referring to line 314)*

    **Response:** Thank you for your suggestions. We have added this reference and information to Section 3.2.

*References:*

*Albinet, A., Minero, C., and Vione, D.: Photochemical generation of reactive species upon irradiation of rainwater: Negligible photoactivity of dissolved organic matter, Sci. Total Environ., 408, 3367–3373, https://doi.org/10.1016/j.scitotenv.2010.04.011, 2010.*

*Anastasio, C. and McGregor, K. G.: Chemistry of fog waters in California's Central Valley: 1. In situ photoformation of hydroxyl radical and singlet molecular oxygen, Atmos. Environ., 35, 1079–1089, https://doi.org/10.1016/S1352-2310(00)00281-8, 2001.*

*Bikkina, S. and Sarin, M.: Brown Carbon in the Continental Outflow to the North Indian Ocean, Env. Sci Process Impacts, 21, 970, 2019.*

*Bogler, S., Daellenbach, K. R., Bell, D. M., Prévôt, A. S. H., El Haddad, I., and Borduas-Dedekind, N.: Singlet Oxygen Seasonality in Aqueous PM10 is Driven by Biomass Burning and Anthropogenic Secondary Organic Aerosol, Environ. Sci. Technol., 56, 15389–15397, https://doi.org/10.1021/acs.est.2c04554, 2022.*

Chen, K., Raeofy, N., Lum, M., Mayorga, R., Woods, M., Bahreini, R., Zhang, H., and Lin, Y.-H.: Solvent effects on chemical composition and optical properties of extracted secondary brown carbon constituents, Aerosol Sci. Technol., 56, 917–930, https://doi.org/10.1080/02786826.2022.2100734, 2022.

Cote, C. D., Schneider, S. R., Lyu, M., Gao, S., Gan, L., Holod, A. J., Chou, T. H. H., and Styler, S. A.: Photochemical Production of Singlet Oxygen by Urban Road Dust, Environ. Sci. Technol. Lett., 5, 92–97, https://doi.org/10.1021/acs.estlett.7b00533, 2018.

Di Lorenzo, R. A., Washenfelder, R. A., Attwood, A. R., Guo, H., Xu, L., Ng, N. L., Weber, R. J., Baumann, K., Edgerton, E., and Young, C. J.: Molecular-Size-Separated Brown Carbon Absorption for Biomass-Burning Aerosol at Multiple Field Sites, Env. Sci Technol, 51, 3128, 2017.

Fan, S. and Li, Y.: The impacts of marine-emitted halogens on OH radicals in East Asia during summer, Atmospheric Chem. Phys., 22, 7331–7351, https://doi.org/10.5194/acp-22-7331-2022, 2022.

Fang, T., Hwang, B. C. H., Kapur, S., Hopstock, K. S., Wei, J., Nguyen, V., Nizkorodov, S. A., and Shiraiwa, M.: Wildfire particulate matter as a source of environmentally persistent free radicals and reactive oxygen species, Environ. Sci. Atmospheres, 3, 581–594, https://doi.org/10.1039/D2EA00170E, 2023.

Faust, B. C. and Allen, J. M.: Aqueous-phase photochemical sources of peroxyl radicals and singlet molecular oxygen in clouds and fog, J. Geophys. Res. Atmospheres, 97, 12913–12926, https://doi.org/10.1029/92JD00843, 1992.

Fleming, L. T., Lin, P., Roberts, J. M., Selimovic, V., Yokelson, R., Laskin, J., Laskin, A., and Nizkorodov, S. A.: Molecular Composition and Photochemical Lifetimes of Brown Carbon Chromophores in Biomass Burning Organic Aerosol, Atmos Chem Phys, 20, 1105, 2020.

Forrister, H., Liu, J., Scheuer, E., Dibb, J., Ziemba, L., Thornhill, K. L., Anderson, B., Diskin, G., Perring, A. E., Schwarz, J. P., Campuzano-Jost, P., Day, D. A., Palm, B. B., Jimenez, J. L., Nenes, A., and Weber, R. J.: Evolution of Brown Carbon in Wildfire Plumes, Geophys Res Lett, 42, 4623, 2015.

Hems, R. F., Schnitzler, E. G., Liu-Kang, C., Cappa, C. D., and Abbatt, J. P. D.: Aging of Atmospheric Brown Carbon Aerosol, ACS Earth Space Chem., 5, 722–748, https://doi.org/10.1021/acsearthspacechem.0c00346, 2021.

Hong, J., Liu, J., Wang, L., Kong, S., Tong, C., Qin, J., Chen, L., Sui, Y., and Li, B.: Characterization of reactive photoinduced species in rainwater, Environ. Sci. Pollut. Res., 25, 36368–36380, https://doi.org/10.1007/s11356-018-3499-4, 2018.

Jiang, W., Ma, L., Niedek, C., Anastasio, C., and Zhang, Q.: Chemical and Light-Absorption Properties of Water-Soluble Organic Aerosols in Northern California and Photooxidant Production by Brown Carbon Components, ACS Earth Space Chem., https://doi.org/10.1021/acsearthspacechem.3c00022, 2023.

Kaur, R., Labins, J. R., Helbock, S. S., Jiang, W., Bein, K. J., Zhang, Q., and Anastasio, C.: Photooxidants from brown carbon and other chromophores in illuminated particle extracts, Atmospheric Chem. Phys., 19, 6579–6594, https://doi.org/10.5194/acp-19-6579-2019, 2019.

Laskin, J., Laskin, A., Nizkorodov, S. A., Roach, P., Eckert, P., Gilles, M. K., Wang, B., Lee, H. J., and Hu, Q.: Molecular Selectivity of Brown Carbon Chromophores, Env. Sci Technol, 48, 12047, 2014.

Lee, A. K. Y., Willis, M. D., Healy, R. M., Wang, J. M., Jeong, C.-H., Wenger, J. C., Evans, G. J., and Abbatt, J. P. D.: Single-particle characterization of biomass burning organic aerosol (BBOA): evidence for non-uniform mixing of high molecular weight organics and potassium., Atmospheric Chem. Phys., 16, 5561–5572, https://doi.org/10.5194/acp-16-5561-2016, 2016.

Lee, H. J. (Julie), Aiona, P. K., Laskin, A., Laskin, J., and Nizkorodov, S. A.: Effect of Solar Radiation on the Optical Properties and Molecular Composition of Laboratory Proxies of Atmospheric Brown Carbon, Environ. Sci. Technol., 48, 10217–10226, https://doi.org/10.1021/es502515r, 2014.

Leresche, F., Salazar, J. R., Pfotenhauer, D. J., Hannigan, M. P., Majestic, B. J., and Rosario-Ortiz, F. L.: Photochemical Aging of Atmospheric Particulate Matter in the Aqueous Phase, Environ. Sci. Technol., https://doi.org/10.1021/acs.est.1c00978, 2021.

Lyu, Y., Lam, Y. H., Li, Y., Borduas-Dedekind, N., and Nah, T.: Efficient production of singlet oxygen and organic triplet excited states in aqueous $PM_{2.5}$ in Hong Kong, South China, EGUsphere, 1–28, https://doi.org/10.5194/egusphere-2023-739, 2023.

Ma, L., Worland, R., Tran, T., and Anastasio, C.: Evaluation of Probes to Measure Oxidizing Organic Triplet Excited States in Aerosol Liquid Water, Environ. Sci. Technol., https://doi.org/10.1021/acs.est.2c09672, 2023a.

Ma, L., Worland, R., Jiang, W., Niedek, C., Guzman, C., Bein, K. J., Zhang, Q., and Anastasio, C.: Predicting photooxidant concentrations in aerosol liquid water based on laboratory extracts of ambient particles, EGUsphere, 1–36, https://doi.org/10.5194/egusphere-2023-566, 2023b.

Manfrin, A., Nizkorodov, S. A., Malecha, K. T., Getzinger, G. J., McNeill, K., and Borduas-Dedekind, N.: Reactive Oxygen Species Production from Secondary Organic Aerosols: The Importance of Singlet Oxygen, Environ. Sci. Technol., 53, 8553–8562, https://doi.org/10.1021/acs.est.9b01609, 2019.

Martin, R. V., Jacob, D. J., Yantosca, R. M., Chin, M., and Ginoux, P.: Global and regional decreases in tropospheric oxidants from photochemical effects of aerosols, J. Geophys. Res. Atmospheres, 108, https://doi.org/10.1029/2002JD002622, 2003.

McNeill, K. and Canonica, S.: Triplet State Dissolved Organic Matter in Aquatic Photochemistry: Reaction Mechanisms, Substrate Scope, and Photophysical Properties, Env. Sci Process Impacts, 18, 1381, 2016.

Ossola, R., Jönsson, O. M., Moor, K., and McNeill, K.: Singlet Oxygen Quantum Yields in Environmental Waters, Chem. Rev., 121, 4100–4146, https://doi.org/10.1021/acs.chemrev.0c00781, 2021.

*Pfannerstill, E. Y., Reijrink, N. G., Edtbauer, A., Ringsdorf, A., Zannoni, N., Araújo, A., Ditas, F., Holanda, B. A., Sá, M. O., Tsokankunku, A., Walter, D., Wolff, S., Lavrič, J. V., Pöhlker, C., Sörgel, M., and Williams, J.: Total OH reactivity over the Amazon rainforest: variability with temperature, wind, rain, altitude, time of day, season, and an overall budget closure, Atmospheric Chem. Phys., 21, 6231–6256, https://doi.org/10.5194/acp-21-6231-2021, 2021.*

*Pimlott, M. A., Pope, R. J., Kerridge, B. J., Latter, B. G., Knappett, D. S., Heard, D. E., Ventress, L. J., Siddans, R., Feng, W., and Chipperfield, M. P.: Investigating the global OH radical distribution using steady-state approximations and satellite data, Atmospheric Chem. Phys., 22, 10467–10488, https://doi.org/10.5194/acp-22-10467-2022, 2022.*

*Thames, A. B., Brune, W. H., Miller, D. O., Allen, H. M., Apel, E. C., Blake, D. R., Bui, T. P., Commane, R., Crounse, J. D., Daube, B. C., Diskin, G. S., DiGangi, J. P., Elkins, J. W., Hall, S. R., Hanisco, T. F., Hannun, R. A., Hintsa, E., Hornbrook, R. S., Kim, M. J., McKain, K., Moore, F. L., Nicely, J. M., Peischl, J., Ryerson, T. B., St. Clair, J. M., Sweeney, C., Teng, A., Thompson, C. R., Ullmann, K., Wennberg, P. O., and Wolfe, G. M.: Missing OH reactivity in the global marine boundary layer, Atmospheric Chem. Phys., 20, 4013–4029, https://doi.org/10.5194/acp-20-4013-2020, 2020.*

**Response References**

Appiani, E., Ossola, R., Latch, D. E., Erickson, P. R. and McNeill, K.: Aqueous singlet oxygen reaction kinetics of furfuryl alcohol: effect of temperature, pH, and salt content, Environ. Sci.: Processes Impacts, 19(4), 507–516, doi:10.1039/C6EM00646A, 2017.

Jiang, W., Ma, L., Niedek, C., Anastasio, C. and Zhang, Q.: Chemical and Light-Absorption Properties of Water-Soluble Organic Aerosols in Northern California and Photooxidant Production by Brown Carbon Components, ACS Earth Space Chem., doi:10.1021/acsearthspacechem.3c00022, 2023.

Kaur, R., Labins, J. R., Helbock, S. S., Jiang, W., Bein, K. J., Zhang, Q. and Anastasio, C.: Photooxidants from brown carbon and other chromophores in illuminated particle extracts, Atmos. Chem. Phys., 19(9), 6579–6594, doi:10.5194/acp-19-6579-2019, 2019.

Ma, L., Worland, R., Jiang, W., Niedek, C., Guzman, C., Bein, K. J., Zhang, Q. and Anastasio, C.: Predicting photooxidant concentrations in aerosol liquid water based on laboratory extracts of ambient particles, Atmos. Chem. Phys., 23(15), 8805–8821, doi:10.5194/acp-23-8805-2023, 2023.

McFall, A. S., Johnson, A. W. and Anastasio, C.: Air-Water Partitioning of Biomass-Burning Phenols and the Effects of Temperature and Salinity., Environ. Sci. Technol., 54(7), 3823–3830, doi:10.1021/acs.est.9b06443, 2020.

Schmitt, M., Erickson, P. R. and McNeill, K.: Triplet-State Dissolved Organic Matter Quantum Yields and Lifetimes from Direct Observation of Aromatic Amine Oxidation., Environ. Sci. Technol., 51(22), 13151–13160, doi:10.1021/acs.est.7b03402, 2017.

---

## Author Response (AR2)

**Response to Editor and Referee #3**

"Seasonal variations in photooxidant formation and light absorption in aqueous extracts of ambient particles" by Lan Ma et. al.

Dear Dr. Sullivan,

Thank you for your review of our manuscript and for soliciting the opinion of a third reviewer. Based on your suggestions, we slashed the abstract to meet the new maximum length requirement. We did not alter our title or conclusions section as these both seem to fit with ACP guidelines.

Below are our responses to the review from Referee #3. Each reviewer comment is listed in *italics* and our response, in plain text, is directly below it. Line numbers in the revised version are different from the original (e.g., in the reviewers' comments) due to changes in the manuscript.

***Anonymous Referee #3***

*I read the revised version of this manuscript after it was reviewed by two reviewers. Reviewer 1 requested relatively minor changes. Reviewer 2 suggested that the presented data overlap with the already published results from the same group in Jiang et al. (2023) and Ma et al. (2023). The methods and results described in this paper are indeed linked to the results presented in Jiang et al. (2023) and Ma et al. (2023), especially the latter. However, the authors provided a convincing explanation in the response about how these three manuscripts are connected and how they are different. This is a very large data set resulting from years of work, so it is understandable that the authors are trying to split into more focused, digestible stories. I recommend publishing the revised paper. I have additional minor comments:*

*1. The abstract is unreasonably long, to the extent it had to be split in two paragraphs. While long abstracts have appeared in published in ACP papers, it is not in a general a good practice. I would strongly recommend condensing it to a more reasonable length focusing on the most important message of this paper and omitting less important details.*

Response: We thank the reviewer for their thoughtful comments and for understanding the connection and differences among the three papers. We agree that the abstract is too long and we have rewritten it to be much shorter.

*2. L120: I would mention that this work builds on the previous study by Kaur et al. (2019) since these data are used in the figures*

Response: Thank you for your suggestion. We have added this information.

*3. L143: the unit for hour is "h" not "hr" (it is used correctly in most other places in the text)*

Response: Thank you for pointing this out. We have corrected it.

*4. L358 and Figure S13: is there a physical reason to expect a linear relationship here? And regardless of the model, should no the fitting curve pass through zero at zero PM/water ratio?*

Response: We think you are referring to Figure S14 instead of Figure S13 here. The three samples that we studied in our past work showed two different behaviors with increasing PM mass/water mass: two showed essentially a constant [•OH], while the third showed a non-linear (second-order polynomial) increase. Thus the samples in Figure S14 might exhibit a range of relationships between [•OH] and extract concentration, making it impossible to know the correct function for the assemblage. Because of this, we chose a simple linear relationship with a non-zero intercept. We chose to fit the intercept because it provides a better fit to the data and because it allows the regression to account for the small but non-zero formation of •OH in our blanks.

*5. L494. "Figure S15 shows the equivalent plot of Figure 3". I think you mean Figure 4. In the caption of figure S15, I would explain how it is different from figure 4.*

Response: Thank you for pointing this out. Yes, it should be Figure 4 and we have corrected it. We also now explain in the caption of Figure S15 how this figure is different from Figure 4 in the main text.

*6. L669 and below: instead of using "compound (X)" why not use the actual names of these compounds? It will not make the text longer but will make it easier to read. You can still use 1,2,3,4,5 in the figure since the structures are shown right there so the labels are not too confusing.*

Response: Thank you for your suggestion. We have added the compound names to most of these instances.

*7. The same reference for Jiang (2023) in ACS Earth Space Chem. is listed twice in the reference section under indexes "a" and "b". Please fix.*

Response: Thank you for pointing this out. We have corrected it.